# Ancestral perinatal obesogen exposure results in a transgenerational thrifty phenotype in mice

Raquel Chamorro-Garcia [1], Carlos Diaz-Castillo [1], Bassem M. Shoucri[1], Heidi Käch[1,5], Ron Leavitt[1], Toshi Shioda[2] & Bruce Blumberg [1,3,4]

Ancestral environmental exposures to non-mutagenic agents can exert effects in unexposed descendants. This transgenerational inheritance has significant implications for understanding disease etiology. Here we show that exposure of F0 mice to the obesogen tributyltin (TBT) throughout pregnancy and lactation predisposes unexposed F4 male descendants to obesity when dietary fat is increased. Analyses of body fat, plasma hormone levels, and visceral white adipose tissue DNA methylome and transcriptome collectively indicate that the F4 obesity is consistent with a leptin resistant, thrifty phenotype. Ancestral TBT exposure induces global changes in DNA methylation and altered expression of metabolism-relevant genes. Analysis of chromatin accessibility in F3 and F4 sperm reveals significant differences between control and TBT groups and significant similarities between F3 and F4 TBT groups that overlap with areas of differential methylation in F4 adipose tissue. Our data suggest that ancestral TBT exposure induces changes in chromatin organization transmissible through meiosis and mitosis.

[1] Department of Developmental and Cell Biology, University of California, 2011 Biological Sciences 3, Irvine, CA 92697-2300, USA. [2] Center for Cancer Research, Massachusetts General Hospital, Bldg 149, 13th Street, Charlestown, MA 02129, USA. [3] Department of Pharmaceutical Sciences, University of California, Irvine, 92697-2300 CA, USA. [4] Department of Biomedical Engineering, University of California, Irvine, 92697-2300 CA, USA. [5]Present address: Department of Environmental Systems Science, ETH, Zurich, 8092, Switzerland. Raquel Chamorro-Garcia and Carlos Diaz-Castillo contributed equally to this work. Correspondence and requests for materials should be addressed to T.S. (email: shioda@helix.mgh.harvard.edu) or to B.B. (email: blumberg@uci.edu)

Obesity is a worldwide health concern in children, adolescents and adults[1]. Major contributing factors leading to obesity are ascribed to increased caloric intake, sedentary lifestyles, or genetic predispositions[2–5]. However, considerable data show that obesity is not simply the result of an imbalance in one's "caloric checkbook"[6]. A recent study of obesity trends based on data from the National Health and Nutrition Examination Study reported that "for a given amount of caloric intake, macronutrient intake or leisure time physical activity, the predicted BMI was up to 2.3 kg m$^{-2}$ higher in 2006 than it was in 1988"[7]. The Dutch Hunger Winter studies showed that children exposed to extreme caloric restriction in utero during the first trimester of pregnancy were predisposed to obesity later in life[8]. Therefore, disturbances in the prenatal environment can cause permanent metabolic consequences - the so-called "thrifty phenotype"[9]. Analysis of 24 animal populations from 8 vertebrate species (domestic dogs and cats, laboratory rats, mice and four species of primates, and feral rats living in cities) showed pronounced increases in obesity in recent decades[10]. These trends suggest that factors beyond caloric intake and energy expenditure are important for developing obesity.

Although positive energy balance is an important cause of obesity, emerging research links early life exposure to endocrine-disrupting chemicals (EDCs) to the obesity epidemic[11–13]. The Endocrine Society defines EDCs as "exogenous chemicals, or mixtures of chemicals, that interfere with any aspect of hormone action"[14]. Obesogens are a subset of EDCs that promote adiposity by increasing fat cell number and/or size, or by interfering with

hormonal regulation of metabolism, appetite, and satiety[11–13]. We[15] and others[16] showed that the obesogens tributyltin (TBT) and triphenyltin (TPT) activate peroxisome proliferator activated receptor gamma (PPARγ), the master regulator of adipogenesis[17], and its heterodimeric partner the retinoid X receptor (RXR). In vitro studies using murine 3T3-L1 preadipocytes and human and mouse mesenchymal stem cells (MSCs) showed that nanomolar levels of TBT promote adipocyte differentiation in a PPARγ-dependent manner[15,18,19]. We showed that exposure of pregnant F0 mice to TBT leads to increased fat storage in white adipose tissue (WAT) and liver, and a shift in the MSC compartment toward the adipogenic lineage and away from the bone lineage[15,18,20]. Significantly, these effects were observed in the F3 generation, which was never exposed to TBT[20]. Transgenerational effects were observed with other EDCs such as bisphenol A (BPA), dichlorodiphenyltrichloroethane (DDT), dibutyl phthalate (DBP), bis(2-ethylhexyl)phthalate (DEHP), and a hydrocarbon mixture (JP-8)[21–23] although the molecular mechanisms underlying these phenomena remain obscure.

Little is known about interactions between obesogen exposure and nutrition, despite the potential relevance of such interactions for the obesity epidemic. The obesogen hypothesis and our previous results suggest that early-life obesogen exposure may alter metabolic set points, predisposing the exposed individuals and their descendants to store more fat[13]. To test this hypothesis, F4 descendants of F0 TBT- or vehicle-exposed animals were subjected to two different metabolic challenges. First, we increased dietary fat content modestly and tracked subsequent changes in

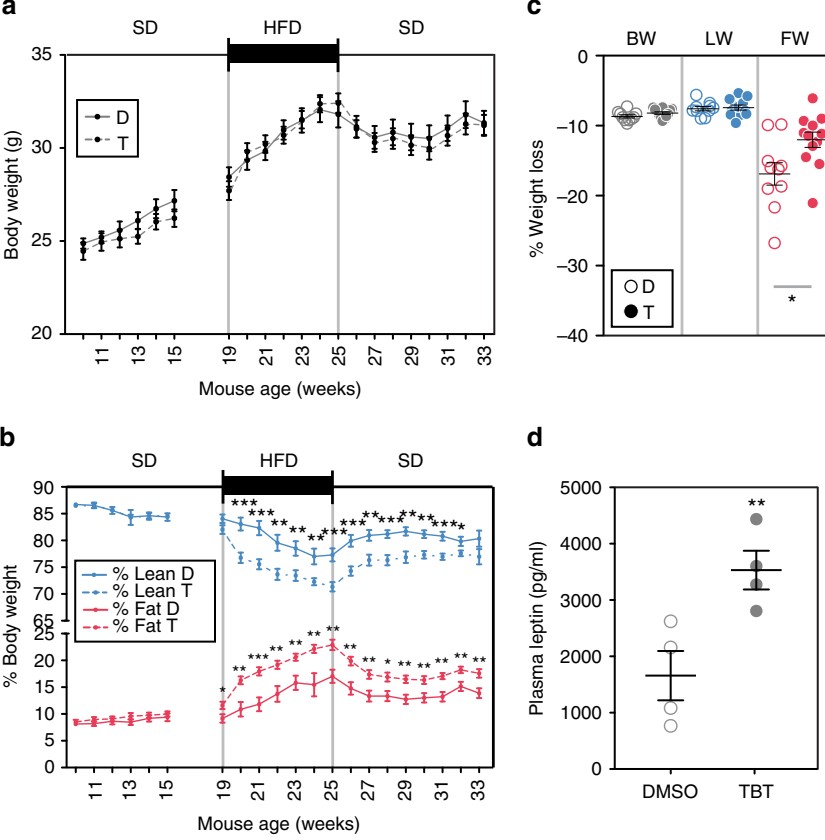

**Fig. 1** Mice ancestrally exposed to TBT develop a leptin resistant "thrifty phenotype". Body weight **a** and relative body composition **b** of F4 DMSO ($n = 10$) and TBT ($n = 12$) males throughout the course of the experiment. **c** Percentage of body weight (BW), lean weight (LW), and fat weight (FW) loss after 16 h of fasting for DMSO ($n = 10$) and TBT ($n = 12$). **d** Plasma levels of leptin in animals at 33 weeks of age (after the diet challenge). D, DMSO; T, TBT. Statistical significance was determined using two-way ANOVA in panels **a**, **b** and one-way ANOVA in panel **c** and **d**. Pair-wise Bonferroni post-tests were used to compare different groups in all panels. Data is presented as mean ± s.e.m. *$p < 0.05$; **$p < 0.01$; ***$p < 0.001$

body weight and composition. Second, we analyzed the effect of fasting on fat mobilization. We found that male animals ancestrally exposed to TBT accumulated more fat than controls on the higher fat diet and maintained this difference after being returned to a normal diet. TBT-group males also lost significantly less fat during periods of fasting than did controls. Whole genome DNA methylome and transcriptomal analysis of gonadal WAT (gWAT) identified blocks of iso-directional changes in DNA methylation (isoDMBs) that were associated with changes in expression of genes associated with metabolic processes, including increased expression of leptin. IsoDMBs were associated with DNA sequence composition which could indicate a link with chromatin organization since DNA sequence composition has been linked with higher order chromatin structure in a recent Hi-C study[24]. Assay for Transposase-Accessible Chromatin with high throughput sequencing (ATAC-seq) revealed changes in chromatin accessibility in F3 and F4 sperm from males ancestrally exposed to TBT compared with controls. Differentially accessible regions in sperm significantly overlap with isoDMBs in gWAT and span regions where genes involved in metabolic processes are located. We infer that TBT exposure can produce alterations in chromatin accessibility, and presumably of nuclear architecture that are transmissible through meiosis and mitosis. Together, these data suggest that rather than targeting the promoters of specific genes, ancestral TBT exposure modifies the metabolic set point of exposed animals by altering chromatin organization resulting in global changes in DNA methylation that elicit a transgenerational thrifty phenotype. These results have broad implications for our understanding of transgenerational phenotypes and how environmental factors may contribute to them.

## Results

**TBT exposure confers a thrifty phenotype on F4 descendants.** Pregnant F0 females were exposed to 50 nM TBT or DMSO vehicle via drinking water given ad libitum throughout pregnancy and lactation. Vehicle- or TBT-exposed animals and their descendants were bred to each other (Supplementary Fig. 1A and Methods). Mice were maintained on low-fat chow (standard diet, SD—13.2% KCal from fat) throughout the experiment (F0–F4). We recorded body weight and fat depot weights in all generations at 8 weeks of age (Supplementary Tables 1, 2). As we previously showed[20] F1 females had increased body weight and peri-renal fat content but this effect largely disappeared in subsequent generations (Supplementary Table 1). F1 males showed modest effects on fat content but increased gWAT content in F2–F4 generations (Supplementary Table 2). We noted decreased weight of the subcutaneous, inguinal adipose depot in males, which has been associated with a decrease in insulin sensitivity both in humans and mice[25].

To evaluate possible interaction between TBT exposure and dietary fat levels, a subset of randomly selected, non-sibling F4 animals was switched to a higher fat diet (HFD—21.2% KCal from fat) at week 19. Animals were maintained on the HFD for 6 weeks, then returned to the SD for 8 weeks until 33 weeks of age (Supplementary Fig. 1B). No significant changes in body weight were observed between vehicle and TBT groups in males or females (Fig. 1a and Supplementary Fig. 2A). However, F4 males descended from TBT-treated animals accumulated significantly more fat than did controls during the first week exposed to the HFD; this trend continued throughout the subsequent 6 weeks (Fig. 1b). When animals were switched back to the SD between weeks 25–33, the TBT group males maintained a significantly higher fraction of body fat than controls (Fig. 1b). We did not detect significant changes in total lean weight or total water

content (Supplementary Fig. 3A); therefore, we infer that increased fat content was at the expense of other body components such as bone minerals that we were not able to measure by MRI, which would explain the lack of differences in total body weight. No significant differences in body composition were observed between TBT or vehicle group females throughout the diet challenge (Supplementary Figs. 2B and 3B).

To evaluate the effect of ancestral TBT exposure on fat mobilization, we examined how animals responded to fasting. One week prior to euthanasia (week 32) animals were subjected to overnight fasting (16 H) with water ad libitum (Supplementary Fig. 1c). Body weight and body composition were measured before and after fasting. Males ancestrally exposed to TBT lost significantly less fat than did control animals (Fig. 1c). We infer that ancestral TBT-exposure led to changes that impaired fat mobilization, which could also contribute to the increased fat content observed in these animals. No significant changes were found in fat mobilization in females or in body weight and lean weight loss in males or females (Fig. 1c and Supplementary Fig. 2C).

Analysis of leptin levels after the diet challenge on week 33 showed that TBT group males had significantly increased levels of plasma leptin, whereas no significant changes were observed in females (Supplementary Table 3). Leptin secretion by WAT is stimulated during meals, which inhibits food intake and increases energy expenditure in a sympathetic nervous system-dependent manner[26]. Alterations of the leptin signaling pathway lead to reduced fat mobilization and increased fat storage[27]. That males from the TBT group had increased plasma leptin levels together with increased fat accumulation and reduced fat mobilization is indicative of a leptin resistant phenotype[27]. Fat storage and fat mobilization are not significantly different between groups in females. To further characterize the leptin resistant phenotype observed in males at the endocrine level, we randomly selected eight males from TBT and control groups and analyzed a panel of seven metabolites related to metabolic diseases; we did not find significant changes in any of the metabolites analyzed (Supplementary Table 4).

**Global analyses of DNA methylome and transcriptome in WAT.** The literature reporting transgenerational experiments identified changes in the DNA methylome of exposed animals, although, how such changes in methylation are transmitted across generations remains uncertain[28,29]. Since WAT plays a central role in both fat accumulation and leptin secretion[27] we examined whether the obese phenotype in males caused by the interaction of F0 TBT exposure and F4 diet challenge was associated with changes in DNA methylation and/or gene expression in the gWAT. We analyzed global DNA methylomes and transcriptomes in gWAT of four randomly selected, non-sibling F4 male animals per group (treatment and control) after they underwent the diet challenge (33 weeks old) using methyl-CpG binding domain deep sequencing (MBD-seq) (Supplementary Fig. 4) and RNA-seq (Supplementary Data 1). MBD-seq reads were mapped to the mouse reference genome sequence (GRCm38/mm10), and divided into consecutive, non-overlapping 100 bp windows. Genomic windows showing statistically significant differential read mapping between the TBT and DMSO groups were denoted as "Differentially Methylated Regions" (DMRs; Supplementary Fig. 4A). "Differentially Expressed Genes" (DEGs) were identified using differential mapping of RNA-seq reads from TBT and DMSO groups to exons in the mm10 annotation (Supplementary Fig. 4C).

Inspection of DMR and DEG trends while varying $p$ values from 0.1 to 0.001 showed that the gWAT genome tended to be

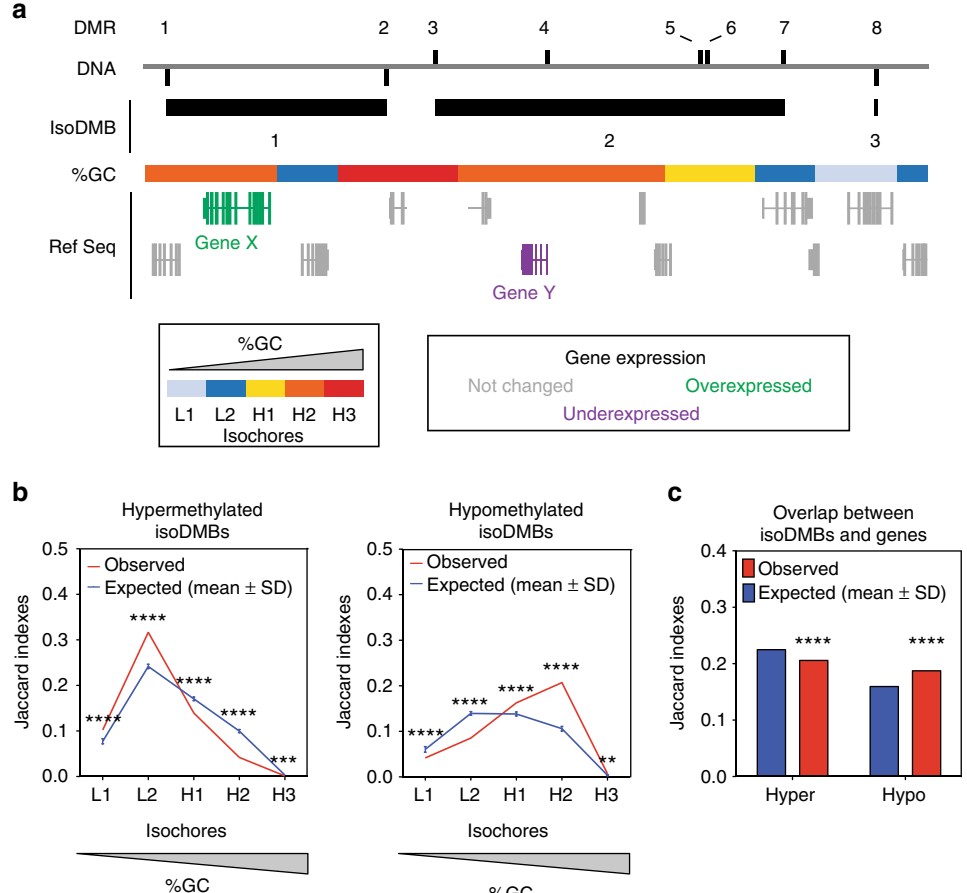

**Fig. 2** TBT-dependent changes in DNA methylation might reflect changes in chromatin organization of F4 male gWAT. **a** Schematic representation of iso-Differentially Methylated Blocks (isoDMBs), and their association with regions with different base composition (%GC-isochores). isoDMBs represent chromosome domains punctuated by iso-directional Differentially Methylated Regions (DMR), i.e., whole genome consecutive, non-overlapping 100 bp windows showing significantly different MBD-seq read coverage between ancestrally TBT- and vehicle-exposed samples. isoDMBs aim to represent cases in which DMRs reflected regional changes in chromatin properties with a potential effect on the expression of genes therein. Hyper Browser was used to calculate the overlap between isoDMBs and each of the five groups defined with different GC content **b** or genes **c**, and to test their statistical significance using Monte Carlo tests[58]. The Jaccard index that measures the similarity between datasets was used to quantify the general overlap between isoDMBs and regions with different base composition or genes (see Methods). Expected Jaccard indexes were calculated after randomly rearranging isoDMBs location 10,000 times respecting isoDMBs, regions with different %GC, and gene length and chromosome assortment. p values represent the number of random simulations showing Jaccard indexes more extreme than observed ones. Data is presented in mean ± s.d. **$p < 0.01$, ***$p < 0.001$, ****$p < 0.0001$

hypermethylated and the transcriptome underexpressed (Supplementary Fig. 4D and Supplementary Data 1). We infer that the DNA methylome and transcriptomal changes in TBT-group F4 male gWAT are coordinated. To explore this possibility we adopted a variation of Monte Carlo-Wilcoxon matched-pairs signed-ranks test (MCW test)[30].

We set a threshold of significance at $p = 0.001$ for DMRs based on their genomic distribution (see Methods). Previous transgenerational analysis focused primarily on DMRs in gene promoters to identify DMRs that persisted through meiosis and mitosis and caused the misregulation of a limited number of genes that explained the transgenerational phenotype[31,32]. This approach is contingent on commonly held assumptions that DNA methylation and demethylation of promoters and regulatory elements results in altered gene expression because it modulates the binding of regulators or the basal transcription machinery to DNA. However, because the fraction of DNA methylation associated with promoters or DNA regulatory motifs is limited, and transcription factor binding to DNA can result in its demethylation, it has recently been suggested that a general function of DNA methylation is the stable silencing of genetic

elements like repetitive DNA or imprinted alleles for genes showing monoallelic expression[33]. Since repetitive DNA and imprinted genes are associated with higher order chromatin structures[33,34] and with transgenerational phenotypes[35–38] it is possible that transgenerational DMRs reflected changes in chromatin organization with a simultaneous effect on multiple genetic elements rather than on individual gene promoters.

To perform a comprehensive analysis of potential associations between TBT-dependent changes in the DNA methylome and transcriptome, we divided genes into three subsets based on their association with DMRs (see Methods). Subset I comprises genes with at least one DMR close to (between −1500 bp and + 500 bp) the transcription start site (TSS). These represent cases where DMRs could affect gene expression by altering DNA binding of short-range regulators or the basal transcription machinery (Supplementary Fig. 5, and Supplementary Data 2, 3). Subset II encompasses genes that overlap or flank at least one DMR irrespective of its distance from the TSS and represents cases where DMRs could affect gene expression by altering DNA binding of chromosome topology modifiers, long- and short-range regulators, or the basal transcription machinery

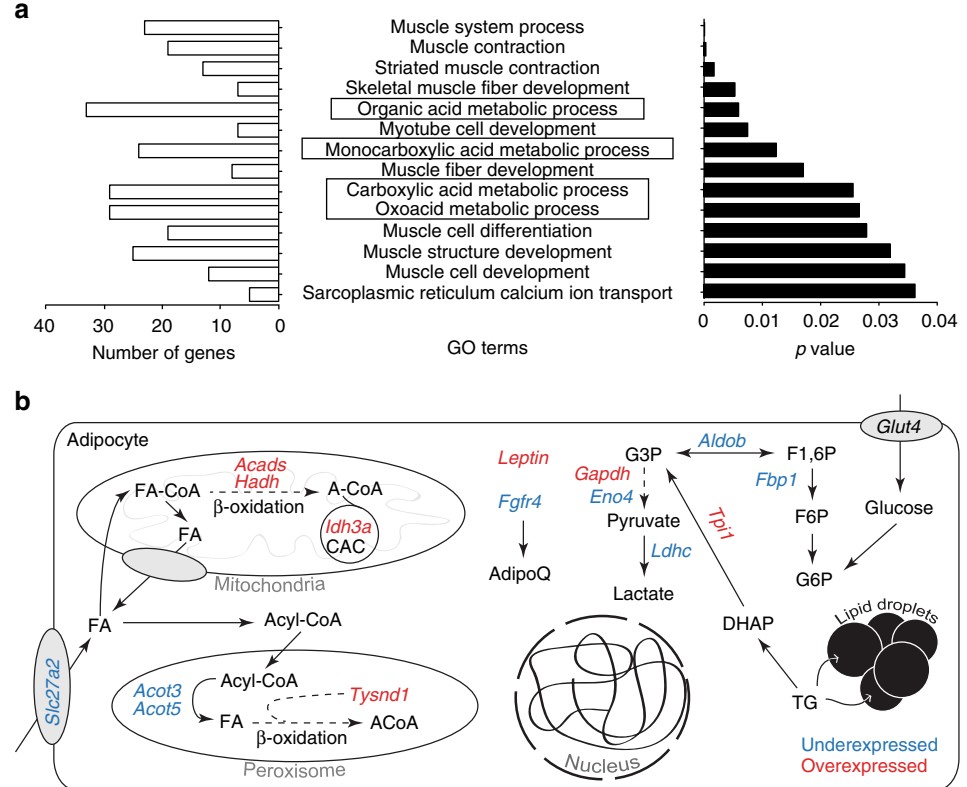

**Fig. 3** Coordinated changes in DNA methylation and gene expression are significantly related with metabolic processes. **a** Gene ontology (GO) enrichment for genes located within isoDMBs that show coherent changes in TBT-dependent DNA methylation and gene expression in F4 male gWAT. Significant enrichment for GO terms were identified using MouseMine, the background population of annotations for mouse genome as the reference set, and Benjamini Hochberg correction for multiple testing to calculate $p$ values according to the hypergeometric distribution (threshold of significance $p = 0.05$)[55]. Right panel represents the $p$ values with most significant enrichments represented on the top of the graph. Left panel shows the number of genes from our dataset that supports significant enrichment for each GO term. GO terms involved in metabolic processes are highlighted with boxes. **b** Schematic representation of the cellular functions of a subset of genes located within isoDMBs showing coherent changes in TBT-dependent DNA methylation and gene expression in F4 male gWAT. Overexpressed and underexpressed genes are represented in red and blue, respectively. The complete list of genes is reported in Supplementary Data 10. ACoA, Acetyl-CoA; CAC, Citric acid cycle; DHAP, dihydroxyacetone phosphate; F1,6P, fructose 1,6 phosphate; F6P, fructose 6 phosphate; FA, fatty acid; G3P, glyceraldehyde 3 phosphate; G6P, glucose 6 phosphate; TG, triglycerides

(Supplementary Fig. 5, and Supplementary Data 2, 3). Subsets I and II rely on DNA methylation changes exerting a direct effect on DNA binding of elements required for gene expression. To explore the possibility that TBT-dependent DMRs and DEGs showed signs of coordination because both were related with changes in higher order chromatin structures, we introduced the concept of iso-Differentially Methylated Blocks (isoDMBs). IsoDMBs are segments of genomic DNA containing iso-directional DMRs (Fig. 2a, and Supplementary Data 4). Subset III encompasses genes located within isoDMBs (Supplementary Fig. 5, and Supplementary Data 3). Although isoDMBs do not preclude the possibility that individual DMRs could have short-range effects on the expression of genes that are close in *cis* or become close spatially through DNA looping, we hypothesize that isoDMBs reflect regional changes in chromatin organization that alter the dynamics of DNA methylation and expression of genes encompassed in them dependent on the ancestral exposure to TBT.

Because variation in DNA methylation was shown to relate with heterogeneity in gene expression[39,40], we independently performed MCW tests for mean transcript abundance and coefficient of variation (CV) for each subset of genes and direction of DNA methylation change in F4 male gWAT, comparing the descendants of the vehicle and TBT groups. Mean transcript abundance is significantly higher than expected by chance for genes in Subset III when they are associated with hypomethylated DMRs (Supplementary Fig. 6 and Supplementary Data 5). Mean transcript abundance is lower than expected by chance for genes in Subsets II and III when they are associated with hypermethylated DMRs (Supplementary Fig. 6 and Supplementary Data 5). Furthermore, transcript abundance CV tends to be significantly higher than expected by chance for genes in Subsets II and III when they are associated with hypermethylated DMRs (Supplementary Fig. 6 and Supplementary Data 5). These trends are consistent with the possibility that changes in gWAT transcriptome of F4 TBT males reflected changes in chromatin organization that modulated DNA accessibility to regulatory and basal transcription machinery. DMRs hypermethylated in TBT animals appear to be associated with repressive chromatin organization. Hypomethylated DMRs would reflect an enhanced accessibility to transcription factors that may cause increased gene expression if factors necessary for expression are present (Supplementary Figs. 7 and 8).

We looked for independent corroborating evidence that might support the results of MCW tests by studying the association of isoDMBs and DEGs defined using a minimum threshold of significance ($p = 0.05$) (Supplementary Fig. 9 and Supplementary Data 6). We reasoned that meaningful isoDMB-DEG coordination would be better exemplified in isoDMBs that harbor more than one differentially expressed gene. If changes in gene

expression were caused by altered chromatin properties reflected by DMRs, then the direction (up or down) of altered expression in multiple DEGs located within the same isoDMB should be consistent. Moreover, they should show the same coordination with the isoDMB direction of change that MCW tests uncovered. Indeed, groups of overexpressed-only DEGs are significantly overrepresented in hypomethylated isoDMBs, and groups of underexpressed-only DEGs are significantly overrepresented in hypermethylated isoDMBs (Supplementary Data 6).

One challenge of long-term transgenerational analyses in mice is to secure sufficient material from each experiment to fulfill initial goals and address questions arising from the characterization of the material generated. Our initial focus in the transgenerational analysis was on DNA methylation and transcriptomal analyses and we used the available material to analyze these endpoints, preventing us from directly addressing the relationship between isoDMBs and chromatin organization in the gWAT experimentally. Therefore, we looked for evidence of this association by examining trends for gene density and base composition within isoDMBs.

Gene density and base composition are genomic properties that are invariant across cell types and have been associated with multiple structural, functional, and evolutionary genomic traits[34,41]. A recent study revealed that the chromatin organization of single oocyte nuclei was only apparent when using base composition bias as guide to identify nuclear components[24]. To analyze the overlap of gene density and base composition in isoDMBs, we used the concept of isochores to divide the genome in five components with different GC content. Isochores are large chromosomal regions with a tendency toward uniformity in base composition that are usually categorized in five classes, L1, L2, H1, H2, and H3, from the most AT-enriched to the most GC-enriched[34]. In F4 TBT males, we found that the extent of hypomethylated isoDMBs covered by either GC-enriched regions (isochores H1, H2, and H3) (Fig. 2b right), or annotated genes (Fig. 2c) is significantly larger than expected by chance. The extent of hypermethylated isoDMBs covered by AT-enriched regions (isochores L1 and L2) is larger than expected by chance (Fig. 2b left) and their extent covered by annotated genes is smaller than expected by chance (Fig. 2c, Supplementary Data 7). GO term enrichment analyses revealed that genes located within GC-enriched isochores are significantly enriched in functional categories related to metabolic processes (Supplementary Data 8). We infer that global changes in DNA methylation resulting from ancestral TBT exposure are likely to be related to structural features that divide the genome into discernible segments, some of which are relevant for metabolism.

Having shown that changes in the transcriptome were more closely associated with isoDMBs than with DMRs in individual genes/promoters, we performed GO enrichment analyses using DEGs found within isoDMBs. We identified 176 DEGs that were overexpressed and located in hypomethylated isoDMBs (47.6%) and 194 that were underexpressed and located in hypermethylated isoDMBs (52.6%) (Supplementary Data 9). These DEGs were significantly enriched in 14 GO terms in two distinctive categories: metabolic processes and muscle biology (Fig. 3a and Supplementary Data 10). 33 DEGs related to metabolic terms are associated with isoDMBs: 17 overexpressed genes were found in hypomethylated isoDMBs and 16 underexpressed genes were found in hypermethylated isoDMBs (Supplementary Figs. 7 and 8 and Supplementary Data 9). These genes participate in metabolic pathways such as β-oxidation, citric acid cycle and glycolysis (Fig. 3 and Supplementary Data 10). Notably, one of the differentially expressed genes we identified is *leptin*, which is associated with a hypomethylated isoDMB (Fig. 4a, b) and GC-enriched DNA (Fig. 4a) and whose mRNA expression was

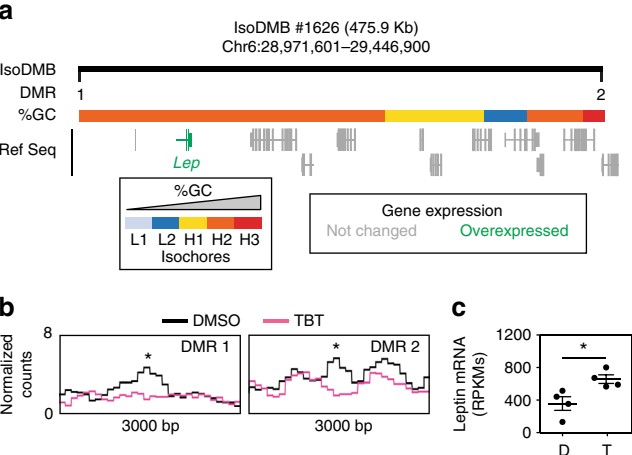

**Fig. 4** Ancestral TBT exposure leads to altered DNA methylation and expression of the leptin gene. **a** Upper panel represents isoDMB #1626 (black bar), and its overlap with regions with different GC content. DMRs punctuating this hypomethylated isoDMB are represented with black vertical bars and numbered (1–2). Overexpressed genes within the isoDMB (Lep, leptin) are represented in green and genes whose expression does not change between DMSO and TBT are represented in gray. **b** Bottom panels show the variation for the mean ($n = 4$) of MBD-seq read coverage for TBT and DMSO samples within 3000 bp regions with the 100 bp DMRs indicated with an asterisk. **c** RPKMs from RNA-seq analysis of *leptin* mRNA expression ($n = 4$). D, DMSO; T, TBT. Statistical significance was determined using R (version 3.3), and Bioconductor (version 3.3) package edgeR (version 3.14)[54]. Data are presented as mean ± s.e.m. *$p < 0.05$

significantly increased in the gWAT in F4 males (Fig. 4c). Analysis of plasma isolated from the same animals, and from the TBT group overall, showed that F4 TBT group males had increased circulating leptin levels (Fig. 1d and Supplementary Table 3). This is a biologically plausible and significant cross-validation of the adequacy of our genomic analysis to infer the existence of TBT-dependent functional modifications that could contribute to the observed obese phenotype.

**Changes in sperm chromatin accessibility.** Integrative analysis of F4 gWAT DNA methylome and transcriptome suggested that TBT-dependent DNA methylome changes could be secondary to, or result from TBT-dependent changes in the chromatin organization in the nucleus. To address if the putative TBT-dependent changes in chromatin organization were related with changes transmissible through meiosis and mitosis, we analyzed chromatin accessibility using ATAC-seq for 24 sperm samples from the descendants of DMSO- and TBT-treated animals: 2 generations (F3, F4) 2 treatments (vehicle vs. TBT) with 6 biological replicates for each. Since a preliminary inspection of ATAC-seq read distribution showed they were not generally organized in sharp peaks, we adopted two analytical strategies to inspect differential ATAC-seq read coverage for broader chromosomal regions. First, we used SICER to define broad areas of the genome with significant enrichment of ATAC-seq reads over background[42]. Genomic regions characterized by base composition uniformity tend to correlate well with chromatin organization[41], and it was observed that base composition-guided analyses for chromatin organization is very successful when dealing with sparse datasets, e.g., single oocyte nuclei[24]. Therefore, we asked whether there was a TBT-dependent differential coverage of ATAC-seq reads in isochores to assess whether TBT-dependent changes in DNA methylation in F4 gWAT showed biased

association with chromosomal regions defined by their base composition uniformity. The results obtained following these two approaches are congruent. However, SICER pools data from biological samples for each treatment reducing the statistical power achieved through biological replication and could make analysis of sparse datasets more susceptible to noise. To avoid this potential bias, we base the following description on isochore-guided ATAC-seq analyses, but note that both approaches give similar results (Supplementary Data 11–21 and Supplementary Figs. 10 and 11).

We started by studying the similarity of F3 and F4 DMSO and TBT samples with regard to their general patterns for chromatin accessibility using Pearson correlation-based hierarchical clustering implemented in chromVar package for the five classes of mouse isochores separately (Supplementary Data 11–14). We observed a significant separation of DMSO and TBT samples when chromVar was guided using H2 isochores (46–53% GC content), and found that the F3 and F4 sperm samples were similar (Fig. 5a and Supplementary Data 13, 14). These results are consistent with the possibility that TBT resulted in a transmissible modification of chromatin organization.

To study TBT-dependent changes in F3 and F4 sperm chromatin accessibility and their potential association with F4 gWAT DNA methylome and the metabolic disruption detected in F4 animals, we identified isochores that significantly differed in their ATAC-seq read coverage. These differentially accessible isochores (DAIs) for F3 and F4 samples were identified independently using MEDIPS and a minimum threshold of significance ($p = 0.05$) (Supplementary Data 11). We refer to DAIs with a significantly higher coverage of ATAC-seq reads in TBT samples than in DMSO samples as accessible DAIs, and to DAIs with a significantly lower coverage of ATAC-seq reads in TBT samples than in DMSO samples as inaccessible DAIs. In both generations, inaccessible DAIs tend to be significantly overrepresented in regions with higher GC content (H2 and H3 isochores), whereas accessible DAIs tend to be significantly overrepresented in regions with lower GC content (L1 and L2 isochores) (Fig. 5b and Supplementary Data 15). This bimodal distribution of isochores with significant differences in chromatin accessibility echoes the bimodal distribution for the overlap between F4 adipose isoDMBs and mouse isochores (Fig. 2c and Supplementary Data 7). These similar distributions could support a mechanistic connection between TBT-dependent changes in sperm chromatin accessibility and adipose DNA methylome. This possibility is underscored by the fact that inaccessible DAIs in sperm significantly overlap with hypomethylated isoDMBs in gWAT, whereas sperm accessible DAIs tend to significantly overlap with hypermethylated isoDMBs in gWAT (Fig. 5c, d and Supplementary Data 16).

Finally, to assess whether TBT-dependent changes in F3 and F4 sperm chromatin accessibility related with the metabolic phenotype observed in F4 animals, we functionally characterized genes spanned by DAIs in F3 and F4 sperm. First, we enquired about the functionality of genes spanned by DAIs found significant and with the same direction of change in both generations, i.e., F3-F4 shared DAIs (Supplementary Data 17). The fraction of F3-F4 shared DAIs is significantly larger than expected by chance (Fig. 6a), but genes spanned by them do not show any significant GO term enrichment (Supplementary Data 17, 18). Second, we looked for significant enrichments of genes spanned by significant DAIs in the same direction for each generation separately. GO terms found enriched for genes spanned by the same type of DAIs in each generation showed a strong similarity (Fig. 6b and Supplementary Data 19). Notably, all GO terms found enriched for both inaccessible F3 DAIs and inaccessible F4 DAIs are associated with metabolic functions

(Fig. 6c and Supplementary Data 19), which connects with the metabolic disruption we observed for F4 animals ancestrally exposed to TBT.

## Discussion

Here we showed that ancestral exposure to the obesogen TBT stimulated fat storage later in life. F4 male descendants of pregnant F0 mice exposed to a low dose of TBT (below the established no observed adverse effect level, NOAEL) via the drinking water showed increased fat mass as adults and gained fat more quickly than controls when switched to a higher fat diet (from 13.2% to 21.2%) (Fig. 1a, b). We also found that fat mobilization during fasting was significantly hampered in F4 TBT-group males (Fig. 1c). Increased fat storage and resistance to fat mobilization during fasting are both hallmarks of Barker's "thrifty phenotype"[43].

Global profiling of the DNA methylome of gWAT from F4 males showed significantly different DNA methylation profiles between animals ancestrally exposed to TBT and controls (Supplementary Fig. 4). We identified regions of genomic DNA where differentially methylated blocks of DNA were all hypo- or hypermethylated and denote these as isoDMBs. IsoDMB-guided analysis of differential gene expression identified hypomethylated isoDMBs containing genes that were both overexpressed and associated with metabolic processes (Figs. 2 and 3 and Supplementary Data 10). We infer that increased expression of these genes is associated with increased chromatin accessibility in these areas. Notably, the leptin gene is overexpressed in gWAT and is located within a hypomethylated isoDMB (Fig. 4). These animals exhibit significantly increased serum levels of leptin compared with controls after the higher fat diet challenge (Fig. 1d and Supplementary Table 3). Despite increased plasma leptin in the TBT group animals, other plasma markers such as insulin and glucagon were unchanged between groups, suggesting that the gWAT is not responding to the increased leptin levels and is, therefore, dysfunctional (Supplementary Table 4)[44]. We also observed altered mRNA expression levels of genes that participate in pathways involved in fatty acid metabolism such as β-oxidation, citric acid cycle and glycolysis (Fig. 3 and Supplementary Data 10). Although β-oxidation (fatty acid catabolism) genes are overexpressed (Fig. 3), genes participating in upstream pathways (glycolysis or fatty acid transport) are underexpressed. Therefore, the increased levels of genes involved in β-oxidation may be a compensatory mechanism in a dysfunctional adipocyte[44]. Altogether, these results demonstrate that ancestral exposure to TBT alters both lipogenic and lipolytic pathways, which is reflected by increased fat storage and decreased fat mobilization. The observed phenotypes fit well with known outcomes of leptin resistance[27].

We uncovered an extensive coordination of DNA methylation and gene expression by adopting a wider view than typical studies of DNA methylation changes in gene promoters. The isoDMB concept connects TBT-dependent changes in DNA methylation with chromatin organization. The study of nuclear architecture, its developmental and evolutionary dynamics, and its association with the etiology and expression of disease spans now more than a century[34,45–47]. Despite diverse methodological approaches from crude microscopic observations to the most powerful current molecular techniques, many studies converged on the identification of a few major genomic components, e.g., euchromatin/heterochromatin, GC/AT-enriched DNA, gene-rich/-poor DNA, early/late-replicating DNA, or components A/B[34]. Principles that differentiate specific regions of the genome appear to coalesce in two major genomic components that are recapitulated here by opposite biases for gene density and base composition for

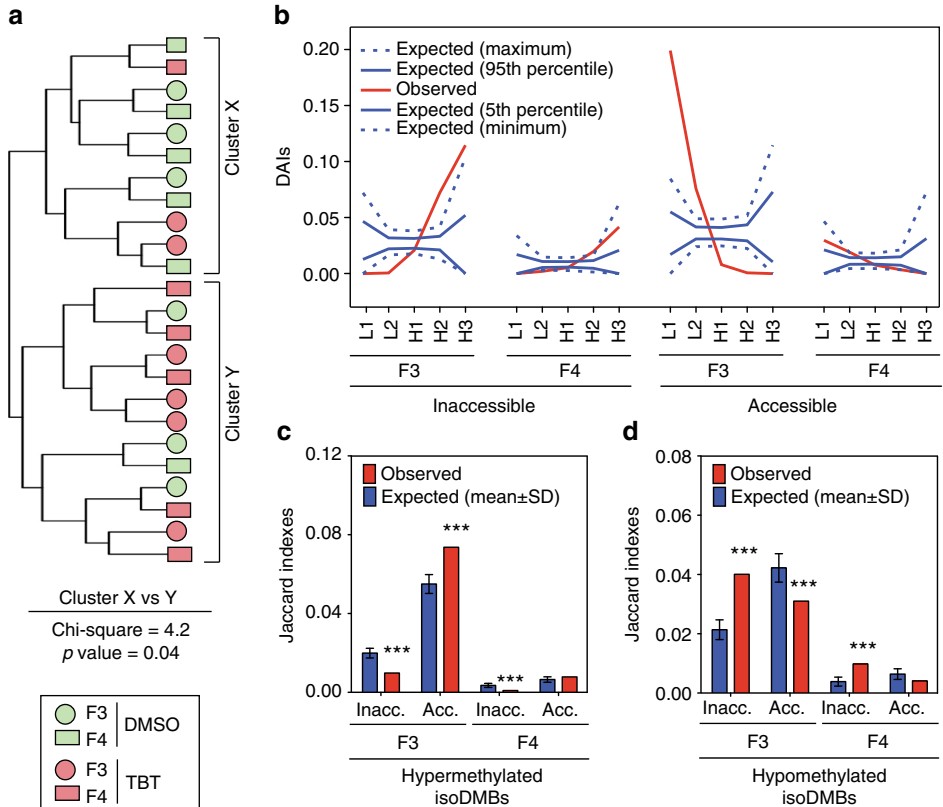

**Fig. 5** Ancestral TBT exposure leads to altered sperm chromatin accessibility. ATAC-seq was used to compare chromatin accessibility patterns ($n = 6$ per treatment group and generation). **a** Hierarchical clustering of DMSO and TBT sperm samples using their chromatin accessibility similarities obtained using chromVar guided by H2 isochores. Chi-square tests were used to compare the assortment of DMSO and TBT, or F3 and F4 samples with regard to the two main clusters (*X* and *Y*), and subclusters within each cluster (see Methods and Supplementary Data 13). **b** Distribution of differentially accessible isochores (DAIs) defined using MEDIPS (see Methods for further details) in F3 and F4 generations. DAIs for which ATAC-seq read coverage was larger in TBT than in DMSO samples were deemed as accessible DAIs (Acc.), whereas DAIs for which ATAC-seq read coverage was larger in DMSO than in TBT samples were deemed as inaccessible DAIs (Inacc.). Observed DAI fractions outlying the area defined by 5th and 95th percentiles lines are considered significant ($p < 0.05$), whereas observed DAI fractions outlying the area defined by minimum and maximum lines are considered highly significant ($p < 0.0001$). **c**, **d** Overlap between DAIs in F3 sperm and hyper and hypomethylated isoDMBs in F4 adipose tissue. The Jaccard index that measures the similarity between datasets was used to quantify the general overlap between F3 and F4 DAIs and F4 adipose isoDMBs (see Methods). Expected Jaccard indexes were calculated after randomly rearranging isoDMBs location 10,000 times respecting isoDMBs, and DAIs length and chromosome assortment using HyperBrowser. *p* values represent the number of random simulations showing Jaccard indexes more extreme than observed ones. Data is presented as mean ± s.d. **$p < 0.01$, ***$p < 0.001$

hyper and hypomethylated isoDMBs (Fig. 2). Regions with uniform base composition have been associated with chromatin structures such as topological associated domains (TADs)[41]. Our results showing a significant overlap between F4 gWAT isoDMBs and regions with uniform base composition suggest a possible association between TBT-dependent adipose isoDMBs and TADs. TBT-dependent changes in DNA methylation may not be the principal cause for the transgenerational inheritance of the ancestral exposure to TBT. Rather they may be secondary to, or caused by changes in chromatin organization. The incipient association we detected between TBT-dependent changes in sperm chromatin accessibility and gWAT DNA methylation (Fig. 5) is consistent with this hypothesis and suggests that TBT-dependent alterations of chromatin organization are transmissible through meiosis and mitosis. The number of chromosomal regions showing similar changes in chromatin accessibility in F3 and F4 sperm is relatively low. However, the functional overlap for genes spanned independently by F3 and F4 DAIs suggests that TBT-dependent changes to chromatin organization rely more on changes to chromosomal regions with similar characteristics than to specific changes in a small number of fixed locations. Further studies specifically designed to analyze the nuclear architecture of

somatic and germ cells will be required to address how TBT results in a transmissible alteration of the nuclear architecture. It is tempting to speculate that TBT-dependent changes in chromatin organization in sperm might alter genomic accessibility to epigenetic writer proteins such as DNA methyltransferases during post-fertilization development, resulting in significant changes in the epigenomes of somatic cells such as adipocytes. In this model, enhanced accessibility would favor DNA methylation whereas reduced accessibility would limit DNA methylation, in accord with our observations. Our study also demonstrates that ancestral TBT-induced changes in epigenetic marks observed in sperm may not necessarily be copied directly to the genomes of the phenotype-associated somatic cells. Instead, epimutations in the germline might affect tissue- and sex-specific epigenetic reprogramming in the genomes of somatic cells in various ways and the locations of such epigenetic effects in somatic cells could be predicted by detecting epimutations in sperm.

Considering that metabolic genes are significantly overrepresented in GC-enriched DNA instead of homogeneously distributed across the genome (Supplementary Data 8), one might reasonably expect that factors causing regionalized changes in chromatin properties will result in altered metabolic outcomes,

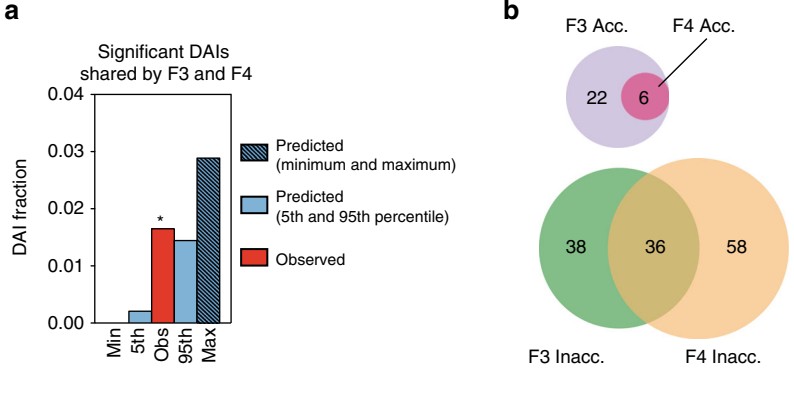

**Fig. 6** TBT-dependent changes in sperm chromatin accessibility preferentially relate with metabolic functions. Chromosomal regions with uniform base composition showing significant differential ATAC-seq read coverage or differentially accessible isochores (DAIs) were defined using MEDIPS (see Methods for further details). DAIs for which ATAC-seq read coverage was larger in TBT than in DMSO samples were deemed as accessible DAIs (Acc.), whereas DAIs for which ATAC-seq read coverage was larger in DMSO than in TBT samples were deemed as inaccessible DAIs (Inacc.). **a** The fraction of significant DAIs with the same direction of change both in F3 and F4 samples i.e., F3-F4 shared DAIs was larger than expected by chance. Data is presented as mean ± s.e.m. *$p < 0.05$. **b** Venn diagrams representing the overlap for GO terms enriched for genes spanned by significant DAIs for each generation separately. GRCm38/mm10 mouse reference genome sequence genes spanned by significant DAIs in the same direction for each generation separately were identified using the tool "Genomic Regions Search" from MouseMine[61]. Significant enrichment for GO terms were identified using MouseMine, the background population of annotations for mouse genome as reference set, and Benjamini Hochberg correction for multiple testing to calculate $p$ values according to the hypergeometric distribution (threshold of significance $p = 0.05$)[61]. **c** List of selected GO terms enriched for genes within significantly inaccessible DAIs in both generations with direct ties to metabolic processes (see Supplementary Data 19 for full list)

such as F4 male obesity in the ancestrally TBT-exposed mice. The observation that multiple, yet very different chemical exposures promoted transgenerationally transmitted effects on obesity[21–23] could indicate that transmissable changes in chromatin structure might be a recurrent theme for the influence of environmental factors on the expression of metabolic phenotypes such as obesity. The observed association between differential gene expression and changes in DNA methylation might reflect functional local heterogeneity in chromosomal accessibility, which could involve structural changes in chromatin. The isoDMB concept may be useful to elucidate such heterogeneity or structural changes as a heuristic guide in future studies.

Although obesity is usually linked to positive energy balance, the presence of many other contributing factors suggests that obesity is much more complex than a simple imbalance in the caloric checkbook. We demonstrated here that obesogen exposure during early development leads to transmissible epigenomic alterations that modify metabolism to favor storage when dietary fat is increased and to inhibit fat mobilization during periods of fasting. That TBT-dependent transgenerational effects on obesity are detectable with the modest increase in fat intake seen here or without further dietary challenges[20] indicates that epigenetic contributions to obesity elicited by obesogen exposure are significant and worthy of further study in human populations.

## Methods

**Animal maintenance and exposure.** Male ($n = 25$) and female ($n = 50$) C57BL/6 J mice (7 weeks of age) were purchased from the Jackson Laboratory (Sacramento, CA). Mice were housed in micro-isolator cages in a temperature-controlled room (21–22 °C) with a 12 h light/dark cycle. Water and food was provided ad libitum unless otherwise indicated. Animals were treated humanely and with regard for alleviation of suffering. All procedures conducted in this study were approved by the Institutional Animal Care and Use Committee of the University of California, Irvine. All tissue harvesting was performed with the dissector blinded to which groups the animals belonged to. At the moment of euthanasia, each mouse was assigned a code, known only to a lab member not involved in the dissection process.

Female C57BL/6 J mice (25 females per treatment group) were randomly assigned to the different treatment groups and exposed via drinking water to 50 nM TBT or 0.1% DMSO vehicle (both diluted in 0.5% carboxymethyl cellulose in water to maximize solubility) for 7 days prior to mating. This TBT concentration was chosen based on our previous study[20] and is five times lower than the established no observed adverse effect level (NOAEL)[48]. Chemicals were administered to the dams throughout pregnancy and lactation. Sires were never exposed to the treatment. No statistically significant differences were observed in the number of pups or the sex ratio per litter among the different groups, thus there were no "litter effects" in our experiments (Supplementary Table 5). The standard procedure in toxicological experiments to eliminate potential "litter effects" is to either use litter as the "$n$" or to choose one animal of each sex per litter to represent the litter (both irrespective of actual litter size). However, this procedure actually introduces a "litter effect" since the number of pups during gestation is well-known to affect subsequent growth trajectory[49], which would be catastrophic for an experiment intended to study metabolic effects of the treatment. Therefore, since the average litter size for both treatment and control was 7 pups, we analyzed only litters that

had six to eight pups to remove this potential confounder. We considered both male and female offspring separately in our analysis. At 8 weeks of age, a subset of animals was euthanized and analyzed for fat content; 1 male and 1 female were randomly selected per litter for analysis. Inguinal, gonadal, peri-renal, interscapular WAT and interscapular brown adipose depots were isolated and weighed. The second subset of mice were used as breeders to generate the second generation of animals (Supplementary Fig. 1A). Therefore, there are no parent-offspring pairs in our analyses. We chose only 1 male and 2 females per litter for breeding. To randomize the breeding process as much as possible, we did not breed siblings and did not breed animals from the same litter with the same male. Unexposed animals were bred to each other and TBT-exposed bred to each other. A similar approach was followed in generations F2-F4. None of the F4 animals subjected to the diet challenge were siblings. Mice were fasted for 4 H prior to euthanasia at 8 weeks or 33 weeks of age (before and after diet challenge, respectively).

**Dietary challenge and fasting analyses**. Animal numbers required for the dietary challenge were estimated using a priori power analyses [G*Power v3.1.5]. Based on our preliminary data, differences in fat content between TBT-exposed and control males when maintained with the HFD are $\geq 23\%$ with SEM within groups of $\leq 10\%$. Hence, setting type I and II errors ($\alpha$ and $\beta$) at 0.05 and the effect size $d = 1.47$, the minimum sample size required for a Power $(1-\beta)$ of 0.8 was calculated to be $\geq 7$ animals per group.

Animals from F0-F4 were maintained on a standard diet (SD–Rodent Diet 20 5053*; PicoLab) throughout the experiment. F4 males and females (avoiding siblings within the same gender) were maintained on a higher fat diet (HFD–Mouse Diet 20 5058*; PicoLab) between weeks 19 and 25. Mice were subsequently returned to the SD for 8 more weeks prior to euthanasia at 33 weeks of age (Supplementary Fig. 1B). Body weight and body composition were measured weekly using EchoMRI™ Whole Body Composition Analyzer, which provides lean, fat and water content information. Total water weight includes free water mainly from the bladder and water contained in lean tissue. For the fasting challenge, animals were fasted for 16 H at week 32. Body weight and body composition were measured before and after fasting and percentage of body weight and fat and lean weight loss was calculated. At week 33 animals were euthanized by isofluorane exposure followed by cardiac exsanguination after 4 H fasting. Blood was drawn into a heparinized syringe and centrifuged for 15 min at 5000 rpm at 4 °C. Plasma was transferred to a clean tube and preserved at −80 °C. Leptin levels from males and females were analyzed using the Mouse Leptin ELISA kit (Crystal Chem). Seven metabolic hormones in plasma were analyzed using Luminex® multiplexing technology (Bio-Plex ProTM Diabetes kit, Bio-Rad, Supplementary Table 4) following manufacture´s protocol in eight non-sibling randomly selected F4 males from each group. Adiponectin levels in plasma were measured using Adiponectin (mouse) EIA Kit (Cayman Chemical). Hormone level changes range from 25% (glucagon) to 220% (ghrelin) with SEM within groups of $\leq 15\%$. Setting the conditions as described above with an effect size $d = 3.47$, the minimum sample size was calculated to be 3 animals per group.

**DNA methylome and transcriptome analyses**. Animal numbers required for methylome and transcriptome analyses was estimated using a priori power analysis [G*Power v3.1.5]. Gene expression changes in tissues are typically $\geq 1.5$ fold with SEM of $\leq 10\%$ (effect size $d = (\mu1-\mu2)/\sigma = 3.92$). Hence, setting type I and II errors ($\alpha$ and $\beta$) at 0.05 and the effect size $d = 3.92$, the minimum sample size required for a Power $(1-\beta)$ of 0.95 was calculated to be $\geq 4$.

Genomic DNA (gDNA) and RNA were isolated from male gWAT using ZR-Duet™ DNA/RNA MiniPrep kit (Zymo Research) of animals sacrificed at 33 weeks of age (after the diet challenge). Nucleic acid samples from 4 randomly selected mice from control or TBT groups were submitted for DNA methylome and transcriptome analyses. Sonicated gDNA (100–200 bp, 500 ng) was subjected to methylated DNA enrichment using the MethylMiner kit (Life Technologies), followed by deep sequencing library construction using the NEBNext Ultra kit (New England Biolabs). Sonicated gDNA without methylation enrichment was used as input control. Following the ENCODE recommendation for ChIP-seq[50], we aimed for 20 million mapped reads (75 nt single) for each gDNA sample. Directional RNA-seq libraries were synthesized from ribosomal RNA-depleted samples using the TruSeq RNA-seq kit (Illumina). To monitor efficiencies of library construction and the sensitivity limit for detecting weakly-expressed RNA transcripts, the ERCC (External RNA Control Consortium) spike-in control RNA mixture was added to samples before depletion of rRNA. Following GTEx (Genome-Tissue Expression) project guidelines[51], we aimed to obtain 50 million mapped reads (paired-end, 75 nt + 75 nt) for each RNA sample.

Statistical evaluation of TBT-dependent methylome and transcriptome variation in F4 male gWAT was performed using R (version 3.3), and Bioconductor (version 3.3) packages MEDIPS (version 1.22)[52], Rsubread (version 1.22)[53] and edgeR (version 3.14)[54]. MEDIPS function MEDIPS.meth was used to estimate the statistical significance of MBD-seq reads differential coverage for GRCm38/mm10 mouse reference genomic 100-bp consecutive, non-overlapping windows with the following parameters: diff.method = edgeR, minRowSum = 10, and diffnorm = quantile. Rsubread function featureCounts was used to assign uniquely mapped RNA-seq reads to GRCm38/mm10 mouse reference genome gene models and count reads with the following parameters: GTF.featureType =

exon, GTF.attrType = gene_id, allowMultiOverlap = TRUE, nthreads = 24, and strandSpecific = 0. edgeR functions cpm and glmQLFTest were used to estimate the number of counts per million per gene model and the statistical significance of RNA-seq reads differential coverage.

DNA methylome and transcriptome integrative analyses consisted of three steps. First, we set a threshold of significance for DNA methylation changes based on their genomic distribution at varying p values. Second, we tested if DNA methylation and transcription changes showed signs of coordination using three different models for the effect DNA methylation variation can have on gene expression. Finally, we functionally characterized genes showing signs of gene expression variation associated with TBT-dependent DNA methylation changes.

We started by setting the threshold of significance for the identification of TBT-dependent DNA methylome changes with regard to their genomic distribution (Supplementary Fig. 4). Differentially Methylated Regions (DMRs) represent genomic 100-bp consecutive, non-overlapping windows for which statistically significant differences in MBD-seq read coverage were observed between TBT- and DMSO-groups at any given p value. Merged Differentially Methylated Regions (mDMRs) result after merging adjacent DMRs with the same direction of change at any given p value. We set the threshold of significance for TBT-dependent DNA methylome changes by evaluating the ratio of mDMR/DMRs, which will approximate whether newly discovered DMRs are independent from already identified DMRs when the significant threshold is changed (Supplementary Fig. 4B). mDMR/DMRs would approach 1 when, upon the relaxation of the significance threshold, newly identified DMRs were mostly unrelated to already identified DMRs. mDMR/DMRs would approach a minimum (equal to the number of chromosomes/number of genomic 100-bp consecutive, non-overlapping windows for all chromosomes, or $8.07 \times 10^{-07}$) when upon the relaxation of the significance threshold newly identified DMRs were mostly adjacent to already found DMRs. We set the threshold of significance for TBT-dependent methylome changes at $p = 0.001$ because this is where mDMR/DMRs ratio starts to decrease consistently, meaning that most of DMRs identified at higher p values tend not to identify new independent DNA methylome changes but to broaden already identified ones (Supplementary Fig. 4B).

To study potential coordination between TBT-dependent changes in the DNA methylome and the transcriptome, we performed a variation of Monte Carlo-Wilcoxon matched-pairs signed-ranks test (MCW test hereinafter)[30]. The MCW test is particularly suitable to study coordinated biases in genome-wide gene expression and DNA methylation because it examines the expression status of all genes annotated in the reference genome in contrast to standard approaches that rely on DEGs defined using arbitrarily chosen thresholds of statistical significance. First, we defined three subsets of genes from GRCm38/mm10 mouse reference genome sequence associated with DMRs using the tool "Genomic Regions Search" from MouseMine[55] (Supplementary Fig. 5). Subset I comprises genes with at least one DMR in the close vicinity (between −1500 bp and +500 bp) of their transcription start site (TSS), representing cases for which DMRs could affect gene expression by altering DNA binding of short-range regulators or the basal transcription machinery (Supplementary Fig. 5 and Supplementary Data 2, 3). Subset II encompasses genes that overlap or flank at least one DMR irrespective of its distance from the TSS and represents cases where DMRs could affect gene expression by altering DNA binding of chromosome topology modifiers, long- and short-range regulators, or the basal transcription machinery (Supplementary Fig. 5, and Supplementary Data 2, 3). Here we introduce the concept of iso-Differentially Methylated Blocks (isoDMBs) as genomic DNA segments comprised by individual DMRs or groups of DMRs with the same direction of change irrespective of their relative distances. isoDMBs aim to represent cases in which DMRs reflected regional changes in chromatin properties with a potential effect on the expression of genes therein (Fig. 2a, and Supplementary Data 4). Subset III encompasses genes located within isoDMBs (Supplementary Fig. 5 and Supplementary Data 3). Second, to exclude from the analyses genes that are not expressed in the gWAT, we calculated transcript abundance mean and coefficient of variation (CV) for TBT and DMSO groups for each gene for which RNA-seq coverage was larger than 0 in at least three TBT replicates and in at least three DMSO replicates. Third, for each gene, transcript abundance measures for DMSO samples were subtracted from TBT samples. Fourth, genes within each subset were ranked in ascending order using the absolute value of TBT-DMSO subtractions, and ranks were signed according to the sign of TBT-DMSO subtractions. Fifth, gene expression bias indexes for each subset of genes were calculated as the sum of signed ranks normalized by the maximum absolute value this sum would have if transcript abundance measures for all genes per subset were biased in the same direction. In this case, gene expression bias indexes range between 1 if transcript abundance means or CVs were larger in TBT than in DMSO for all genes in each subset, and −1 if transcript abundance means or CVs were larger in DMSO than in TBT for all genes in each subset. Finally, the statistical relevance of observed gene expression bias indexes was estimated by repeating the process after randomly permuting subset tags for each subset 10,000 times while respecting their chromosomal assortment (Supplementary Data 5). $p_{upper}$ values represent the fraction of random permutations for which gene expression bias indexes were larger or equal than observed ones, whereas $p_{lower}$ represent the fraction of random permutations for which gene expression bias indexes were lower or equal than observed ones.

The significance of the coordination between TBT-dependent variation in the DNA methylome and the transcriptome was further inspected by studying the

fraction of isoDMBs that harbor more than one Differentially Expressed Genes (DEGs) defined using the most commonly accepted threshold of significance ($p = 0.05$). The fraction of isoDMBs containing at least two DEGs was calculated before and after randomly rearranging DEG tags 10,000 times, while respecting their chromosomal assortment (Supplementary Data). $p_{upper}$ values represent the fraction of random permutations for which the fraction of DEGs within isoDMBs or the fraction of isoDMBs containing at least two DEGs were larger or equal than observed ones, whereas $p_{lower}$ represent the fraction of random permutations for which isoDMBs annotated genes or isochores overlaps were lower or equal than observed ones.

To test whether isoDMBs truly reflect regional changes in chromatin properties, we proceeded to study the overlap of isoDMBs with annotated mouse genes and with isochores (Supplementary Data 7). GRCm38/mm10 mouse gene coordinates were obtained using the Mouse Genome Database (MGB)[56]. Isochores represent regions with a tendency for base composition uniformity[34], and their coordinates for GRCm38/mm10 mouse genome annotation were obtained using IsoFinder[57]. The overlap of isoDMBs and mouse annotated genes coordinates was used as proxy to represent gene density for each isoDMB. The overlap of isoDMBs and mouse isochores was used as proxy to represent base composition biases within each isoDMB. Variation for gene density and base composition were shown to extensively correlate with each other, as well as with several structural, functional, and evolutionary genomic features, such as chromatin compaction, replication timing, or the abundance of repetitive DNA or housekeeping genes[34]. isoDMB overlap with either annotated genes or isochores was estimated using Hyper Browser[58], and the statistical significance for these overlaps was inspected using Monte Carlo simulations. The overall extent to which isoDMBs, annotated genes or isochores overlap was measured using the Jaccard index, which is calculated by dividing the total number of base pairs where isoDMBs overlap with either annotated genes or isochores by the total number of base pairs spanning isoDMBs or annotated genes, or isoDMBs or isochores. Jaccard indexes were calculated before and after randomly rearranging isoDMBs genomic locations using GRCm38/mm10 mouse genome annotation 10,000 times, while respecting the size and chromosomal assortment of isoDMBs, annotated genes, and isochores (Supplementary Data 7). $p_{upper}$ values represent the fraction of random permutations for which isoDMBs and annotated genes or isochores overlaps were larger or equal than observed ones, whereas $p_{lower}$ represent the fraction of random permutations for which isoDMBs and annotated genes or isochores overlaps were lower or equal than observed ones. Observed Jaccard indexes significantly higher than simulated indexes represent cases in which the extent of isoDMBs covered by genes or isochores is larger than expected by chance, whereas observed Jaccard indexes significantly lower than simulated indexes represent cases in which the extent of isoDMBs covered by genes or isochores is smaller than expected by chance.

Literature exists suggesting that in mammalian genomes metabolic genes tend to be GC-enriched[59], but to the best of our knowledge the relative location of metabolic genes with regard to mouse isochores had never been inspected. To explore this issue, we defined two groups of genes upon their location within L or H isochores using MouseMine "Genomic Regions Search" tool[55], and GRCm38/mm10 isochores coordinates obtained using IsoFinder[57]. Significant enrichment for GO terms was identified using MouseMine, the background population of annotations for mouse genome as reference set, and Benjamini Hochberg correction for multiple testing to calculate p values according to the hypergeometric distribution (threshold of significance $p = 0.05$)[55].

To functionally characterize genes with the most coherent TBT-dependent methylome and transcriptome variation, we selected overexpressed DEGs located within hypomethylated isoDMBs and underexpressed DEGs located within hypermethylated isoDMBs in fat tissue (Supplementary Data 8). GO analyses were performed as described above.

**ATAC-seq analyses on sperm samples.** Mouse sperm were isolated from 6 randomly selected, non-sibling males per generation. Epididymis and vas deferens were dissected and sperm was isolated as previously described[21]. Briefly, epididymis and vas deferens were placed on a petri dish containing 2 mL of nucleus isolation medium (NIM) buffer (123.0 mmol/l KCl, 2.6 mmol/l NaCl, 7.8 mmol/l NaH$_2$PO$_4$, 1.4 mmol/l KH$_2$PO$_4$ and 3 mmol/l EDTA (disodium salt)). Sperm was pushed out of the tissue and media was collected and transferred to two clean microcentrifuge tubes and kept on ice. Samples were centrifuged at 13,000xg for 5 min at 4 °C. Supernatant was removed and cells were resuspended in 1 mL of fresh NIM buffer. Collected sperm were frozen in dry ice and stored at −80 °C until use. Microscopic observation did not reveal contamination of epithelial cells.

Although ATAC-seq analyses are recommended to be performed on fresh cells, the sperm material we had access to from the experiment described here had been frozen (see above). ATAC-seq was performed as described by Jung et al.[60] Tagmented sperm DNA was subjected to PCR amplification for Illumina deep sequencing library construction as previously described[61]. To avoid batch effects, we randomly selected one sperm specimen from each of the four treatment groups —namely, F3 DMSO, F3 TBT, F4 DMSO, and F4 TBT—and performed the ATAC-seq reaction for these four specimens as a batch on the same day. We processed six batches (total 24 sperm specimens, 6 specimens for each treatment group) as independently performed experiments. The ATAC-seq libraries were sequenced using Illumina NextSeq 500 sequencer and, although we did not start

with fresh cells, we succeeded in producing abundant ATAC-seq reads with adequate quality to test if ancestral exposure of TBT caused transgenerationally transmissible changes in the higher order chromatin organization of sperm. We obtained approximately 100 million reads per treatment group (17–20 million reads per individual specimen). Raw sequencing data (fastq, 50 nt, single read) were subjected to quality control processing using fastQC and Trim Galore (Babraham Institute) to remove low-quality reads and adapter sequences and aligned to mouse GRCm38/mm10 genome reference sequence using STAR aligner without splicing recognition[62]. From the resulting.bam format aligned reads, uniquely aligned reads were extracted using the markdup function of sambamba[63].

We used the package chromVar to test whether chromatin accessibility was generally different between DMSO and TBT samples[64]. Since chromVar was designed to deal with sparse datasets such as those obtained from single cell analyses, it requires a set of chromosomal regions as guide to focus analyses on those areas where meaningful differences are expected[64]. The sperm genome is highly compacted due to the extensive substitution of histones by protamines during spermiogenesis[65]. Because most loci accessible to the Tn5 transposome will be located within regions that retain histones through spermiogenesis, and the extent of the genome that retains histones and their distribution are still contentious[60,66,67], we decided to guide our chromVar analyses using two different approaches. We separately assessed differentially accessible regions mapped using the package SICER, and also by relating the differentially accessible regions to areas of biased DNA sequence composition, identified by the 5 mouse isochores. SICER permits the identification of broad chromosomal regions, referred to as islands, that have significant enrichment in reads over background[68]. SICER islands were defined using the following parameters: window size = 200 bp, fragment size = 150 bp, threshold for redundancy allowed for treated reads = 1, threshold for redundancy allowed for WT reads = 1, the shift for reads = 75, gap size = 600 bp, Evalue for identification of candidate islands that exhibit clustering = 100, and FDR for significance of differences = 0.05 (Supplementary Data 11).

Sample efficiencies were calculated as the number of reads within SICER islands, or each isochore class divided by the total number of reads in each sample. To compare sample efficiencies for chromosomal regions used as reference spanning different fractions of the genome, efficiency measures were also inspected after being normalized by the cumulative length of SICER islands or each isochore class (Supplementary Fig. 10 and Supplementary Data 12). Samples were filtered if their depth was lower than 1,500 reads, or had efficiency lower than 0.15. chromVar was executed using default settings for background peak generation, and setting the threshold for variability to 1.5. Analyses using isochores L1 and H3 did not fulfill these criteria and were unsuccessful. The similarity of chromatin accessibility between DMSO and TBT samples was inspected using hierarchical clustering based on Pearson correlations calculated after removing highly correlated and low variability annotations, and setting autocorrelations to 0 (Supplementary Data 13). Chi-square tests were used to examine the separation of TBT and DMSO or F3 and F4 samples with regard to the two larger clusters for each dendrogram (clusters $X$ and $Y$ from the top to the bottom). Since for H2 isochore-guided analyses, DMSO and TBT sample separation with regard to clusters $X$ and $Y$ was statistically significant ($p < 0.05$), a second round of Chi-square tests was used to examine the separation of DMSO samples from different generations with regard to the two main subclusters within cluster $X$ (subclusters $X'$ and $X''$ from the top to the bottom of the dendrogram), and the separation of TBT samples from different generations with regard to the two main subclusters within cluster $Y$ (subclusters $Y'$ and $Y''$ from the top to the bottom of the dendrogram) (Supplementary Data 14).

To further characterize chromosomal regions with differential coverage for ATAC-seq reads, we used SICER islands as previously defined and isochores for which ATAC-seq read significant differential coverage was assessed using MEDIPS (Supplementary Data 11). Isochores with significantly different ATAC-seq read coverage or significant differentially accessible isochores (DAIs) were defined using MEDIPS.createROIset to extract ATAC-seq read coverage mapped to mouse isochores, MEDIPS.meth to estimate the significance of the differential coverage for DMSO and TBT samples as previously indicated but without correcting for CpG genomic distribution, and using the most commonly accepted threshold of significance ($p = 0.05$). The significance of the fraction of DAIs for each isochore type, and the fraction of DAIs showing the same direction of change in both generations was estimated by calculating DAI fractions before and after randomly rearranging direction of change tags 10,000 times (Supplementary Datas 15–17). $p_{upper}$ and $p_{lower}$ values represent the fraction of random permutations for which isochore fractions were larger or equal, or lower or equal than observed ones, respectively. The overlap of SICER islands and DAIs between generations, and of each one of them with F4 adipose isoDMBs was estimated using Hyper Browser[64] as previously indicated (Supplementary Data 16 and 20). Genomic regions representing the overlap between SICER islands with the same direction of change in F3 and F4 sperm were defined using the tool "Intersect" from Galaxy[69]. Genes spanned by SICER islands or DAIs, and their significant enrichment for GO terms were inspected using MouseMine tools as previously indicated (Supplementary Data 18, 19, and 21). Venn diagrams were drawn using eulerAPE v3[70].

**Data availability**. Raw and processed data are available at Gene Expression Omnibus (GSE105051). Annotated R code for our analyses is included in Supplementary Methods.

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

## Acknowledgements

We would like to thank all the members from the Blumberg laboratory for their contribution in scientific discussion and Dr. Amanda Janesick (Stanford University) for comments on the manuscript. We are also grateful to Dr. Victor Corces, Emory University, for his technical advice on sperm ATAC-seq. This project was funded by the NIEHS grant 5R01ES023316 to B.B. and T.S.

## Author contributions

R.C.-G. and B.B. designed the experiments reported in this manuscript. R.C.-G., B.M.S., H.K., R.L. and T.S. performed the experiments. R.C.-G., C.D.-C. and T.S. analyzed the data. R.C.-G., C.D.-C., T.S. and B.B. wrote the manuscript.

## Additional information

**Competing interests:** B.B. is a named inventor on U.S. patents related to PPARγ. The remaining authors declare no competing financial interests.

