## [Peer Review File · Nature Communications]

Reviewers' comments:

Reviewer #1 (Remarks to the Author):

This manuscript appears to describe a very well conducted study to investigate the induction and subsequent transgenerational transmission of phenotypes associated with obesity in descendants of pregnant female mice exposed to the endocrine disruptor TBT – or to vehicle only (controls). Phenotypes were detected in F3 generation descendants despite no further exposure after the F0 generation dams, suggesting transgenerational epigenetic transmission of induced aberrations in the genome/epigenome.

The analysis of gene expression (RNA-seq) and methylome (me-DIP) patterns is of particular interest. The authors introduce a new term/concept – “isoDMBs” which are said to “represent genomic DNA segments punctuated by iso-directional DMRs and represent cases in which DMRs reflect regional changes in chromatin properties with a potential effect on the expression of genes therein.” A better explanation of this concept would be useful. Some key details seem to be lacking – e.g. how can anything be said about “chromatin properties” if no analysis of chromatin (other than DNA methylation status) was performed? What is the extent of any locational relationship between a DMR and a differentially expressed gene? Is a DMR that is not located near the TSS or promoter of such a gene considered functionally significant? It may well be that considerations such as these are, in fact, accommodated in the author’s notion of an isoDMB, but if so, this needs to be explained more clearly in this reviewer’s opinion.

Regarding the discussion of the methylome data – It is not clear what constitutes a “congruent change in the DNA methylome and transcriptome.” Also there is discussion of “CG-enriched chromosomal domains” – are these equivalent to or distinct from the CpG islands commonly noted in other studies? The discussion of potential effects at the level of “chromosome segmental structure” is both highly speculative and somewhat vague. Is this segmental structure similar to what others now refer to as topologically associated domains (TADs)? In any case, it is not clear that there is direct evidence for the authors’ suggestion that “changes in DNA methylation related to gene expression that we observed are likely associated with chromosome segmental structure.”

Overall this study has yielded several important observations including well supported documentation of another example of transgenerational epigenetic transmission of defects/alterations in the epigenome follow exposure of a pregnant female to an endocrine disruptor, detailed assessments of the phenotypic effects persisting over at least 3 generations with no additional exposure to the disruptive agent following the initial exposure of the F0 generation pregnant dam, and extensive genomics level analyses of gene expression and the methylomes. Novel relationships between these parameters are suggested, which, with some clarification could represent a useful contribution to the field.

Reviewer #2 (Remarks to the Author):

SUMMARY

The authors assess changes in DNA methylomes and transcriptomes in gonadal adipose tissue of mice ancestrally exposed to tributyltin (TBT). Rather than finding specific DNA methylation changes in regulatory regions, however, the authors identify larger chromosomal segments that are coordinately hypo- or hypermethylated (iso-DMBs) that may alter regional chromatin landscapes and, subsequently, gene expression. Differentially expressed genes within these iso-DMBs appear to be enriched for muscle biology and metabolism, suggesting these chromosomal alterations may be associated with the obese, leptin-resistant phenotype in ancestrally exposed mice.

MAJOR CONCERNS (GENERAL AND BY FIGURE)

- The authors make substantial claims (specific examples: [1] regional changes in chromatin properties are reflected by changes in DNA methylation and [2] TBT alters chromosome segmental structures that then result in changes in global DNA methylation) that would be more convincing if additional evidence was provided. For example, other assessments of chromatin landscape (ex. histone modifications, chromatin accessibility, etc.) should be performed.
- The concept of an iso-DMB is difficult to understand with respect to the mechanism of ancestral TBT exposure. This is not really addressed in the paper. In the discussion the authors include one sentence relevant to this question: "The fact that hyper and hypomethylated isoSMBs tend to show opposite biases for gene density and base composition suggests that TBT-dependent DNA methylation variation could be related to the spatial dynamics of the genome." What does this mean? Additionally, have these domains been independently defined in other studies, for example Hi-C studies? Are these blocks reflective of open and closed chromatin? This could easily be determined with ATAC-seq.
- Lack of rationale:
 - o 1. What was the rationale for looking at fat depot weights in F2-F4 generation animals when there were no significant differences in depot weight in the F1 generation?
 - o 2. Why was the methylome and transcriptome of gonadal fat assessed over other fat depots? Is it known whether different fat pads are more/less susceptible to environmental stimuli such as endocrine disruptors?
- Lack of validation:
 - o The most important validation needs to be on the DNA methylation. The assay used here IPs methylated DNA in a nonlinear fashion (regions with more methylated CpGs density are greatly favored).
 - o Leptin mRNA levels in additional animals (besides ones used for RNA-Seq)?
 - o If circulating/serum leptin is also increased in ancestrally TBT-exposed animals, would leptin mRNA levels increased in other fat depots as well?
 - o Are leptin levels significantly different in ancestrally TBT-exposed females compared to control females??

Figure 1

- Is this the F4 generation? This information should be included in the legend.
- Panels A and B: Body weight, % lean, and % fat mass: Animals in the ancestral TBT exposure group already displayed significantly increased percent body fat before the high fat diet challenge. Is the diet challenge necessary to see a significant difference in percent body fat over time between control and TBT groups? Also, is it known if all fat depots are equally affected by the dietary challenge?
- Panel D: There is a discordance between the figure legend and the panel? The figure legend states that plasma leptin is being measured at 8 and 33 weeks, before and after diet challenge. Main body of text (lines 133-136) makes it seem like these are plasma leptin levels measured in 33 week old animals ONLY that were subject to the HFD challenge.

Figure 2

- If isoDMBs alter the entire chromosomal region, would you expect more genes to be differentially expressed within the isoDMB? The supplementary figures show that there are many genes unchanged within an isoDMB (especially sup Fig. 8). Also there are 2 under expressed genes and 3 overexpressed genes in this figure. Are the depicted regions the best examples?
- It is unclear what the isochore data adds to this story. Does the associated isochore categorization contribute to predictability of isoDMB directionality (hypo- or hypermethylated) and/or gene expression (up- or downregulated)?

Figure 4

- Panel A: Should tick marks be pointing down if this is a hypomethylated iso-DMB associated with overexpression of genes?

Supplementary Figure 8

- Iso-DMB is hypo- or hypermethylated? Make sure figure legend and image depict same directionality

Supplementary Table 1 and 2: Total weight and fraction of fat depots in F1-F4 males and females at 8 weeks of age

- Was this analysis done for the 33 week old animals?
- Is the amount of variability across generations concerning?
- Is the severity of the phenotype diminishing over time?
- Why/how do you explain the significant REDUCTION in inguinal fat mass in the fourth generation? This is not addressed at all in the body of the text

MINOR CONCERNS

- Line 39: one's (need apostrophe) instead of ones
- Line 42: squared symbol should be superscript in 2.3 kg/m²
- Making sure italicize *in utero*, *in vitro*, *in vivo*, *ad libitum* throughout the text (some examples/locations/lines: 43-44, 59, 62, 93, 95, 126, 248)
- Repetitive information (lines 170-185 and lines 435-451) classifying genes into Subsets I-III
- How old were the mice used for the methylome and transcriptome studies? Indicate this in the Methods

Reviewers' comments:

Reviewer #1 (Remarks to the Author):

This manuscript appears to describe a very well conducted study to investigate the induction and subsequent transgenerational transmission of phenotypes associated with obesity in descendants of pregnant female mice exposed to the endocrine disruptor TBT – or to vehicle only (controls). Phenotypes were detected in F3 generation descendants despite no further exposure after the F0 generation dams, suggesting transgenerational epigenetic transmission of induced aberrations in the genome/epigenome.

We thank the reviewer for these positive comments

The analysis of gene expression (RNA-seq) and methylome (me-DIP) patterns is of particular interest. The authors introduce a new term/concept – “isoDMBs” which are said to “represent genomic DNA segments punctuated by iso-directional DMRs and represent cases in which DMRs reflect regional changes in chromatin properties with a potential effect on the expression of genes therein.” A better explanation of this concept would be useful. Some key details seem to be lacking – e.g.

1. how can anything be said about “chromatin properties” if no analysis of chromatin (other than DNA methylation status) was performed?

We clarified the definition of the isoDMB concept to link it more closely with well-described existing expressions, such as higher order chromatin structures/organization.¹ The reviewer is correct that we have not specifically demonstrated changes in chromatin structure. What we mean is that isoDMBs represent segments of genomic DNA containing iso-directional, differentially methylated regions that are associated with altered expression of one or more genes within the isoDMB. We propose that these changes in gene expression may involve altered accessibility to DNA and/or changes in other epigenetic marks that reflect altered chromatin structure. Our initial focus in the transgenerational analysis was on DNA methylation and transcriptomal analyses and we used the available material to analyze these endpoints. Therefore, when it became clear that DNA methylation was not changing at specific promoters, but rather in isoDMBs, we did not have any additional material available, particularly freshly-prepared adipose nuclei which are required to perform ATAC-seq or Hi-C and related analyses. We attempted to address the point raised by this reviewer by conducting ATAC-seq analyses of sperm (the only remaining samples from this experiment) from F3 and F4 males ancestrally exposed to vehicle or TBT. ATAC-seq analysis revealed that chromatin accessibility patterns differ between TBT and DMSO samples when guided by DNA sequence composition (using isochores as a proxy for sequence composition). We believe that these results are relevant because they establish an independent connection between ancestral TBT treatment and altered chromatin accessibility that may be indicative of changes in higher order chromatin organization that were transmissible through meiosis and mitosis. This is discussed in the text (lines 305-330).

What is the extent of any locational relationship between a DMR and a differentially expressed gene?

We did not observe strong local correlations between the methylation state of individual DMRs and the expression of adjacent genes. However, once we took a higher level view and asked whether gene expression was altered within regions where DNA methylation was in the same direction, isoDMBs, we observed conservation between these isoDMBs and gene

expression.

2. Is a DMR that is not located near the TSS or promoter of such a gene considered functionally significant? It may well be that considerations such as these are, in fact, accommodated in the author's notion of an isoDMB, but if so, this needs to be explained more clearly in this reviewer's opinion.

IsoDMBs are not closely related to the TSS or promoters of differentially expressed genes. Since we observed that isoDMBs are associated with altered expression of metabolism-relevant genes within the isoDMBs, we infer that DMRs not located near the TSS/promoters have functional significance. While we do not exclude the possibility that methylation or demethylation events near the TSS or promoters of genes whose expression was altered in the TBT group vs controls could occur, we did not observe this. Instead, genome-wide DMR analysis revealed well-localized long-range effects of the ancestral TBT exposure. The isoDMB concept is the outcome of our attempt to link differentially methylated regions with differential gene expression and the relevance of isoDMBs to base composition and metabolic gene expression seems significant. Therefore, isoDMBs have at least a heuristic value in the analysis of differential gene expression, whether or not they reflect actual long-range changes in the chromatin structure (although we believe that they do and ATAC-seq support this view).

Regarding the discussion of the methylome data –

3. It is not clear what constitutes a “congruent change in the DNA methylome and transcriptome.”

We rewrote this sentence to reflect the commonly observed tendency that transcriptional activities are relatively suppressed at heavily methylated genomic DNA regions whereas decreased DNA methylation is often associated with increased mRNA levels (lines 344-349).

4. Also there is discussion of “CG-enriched chromosomal domains” – are these equivalent to or distinct from the CpG islands commonly noted in other studies?

We thank the reviewer for pointing out this potential confusion. We rewrote this statement to indicate that what we meant was that such chromosomal domains were enriched in GC content compared with AT content, rather than actual or potential CpG islands (lines 386). We also replaced "CG" with "GC" when referring to DNA base composition areas to avoid further confusion.

5. The discussion of potential effects at the level of “chromosome segmental structure” is both highly speculative and somewhat vague. Is this segmental structure similar to what others now refer to as topologically associated domains (TADs)?

We agree that there is not yet direct evidence (beyond our new ATAC-seq data) to demonstrate altered chromosome segmental structure. Based on the linkage of isoDMBs and base composition we observed and between TADs and base composition observed by others,² we hypothesize that there could be potential overlap between TADs and isoDMBs. However, because we do not have data for precise locations of TADs in the adipose tissue in our experiments, exactly how isoDMBs are associated with TADs remains to be elucidated in future studies. Indeed, this topic represented 2 out of 3 aims proposed in a new 5-year grant application we just submitted.

6. In any case, it is not clear that there is direct evidence for the authors'

suggestion that “changes in DNA methylation related to gene expression that we observed are likely associated with chromosome segmental structure.”

We agree that unambiguous, direct evidence is not currently available for the statement we made and commented on by the reviewer. We have revised this to “It is tempting to speculate that the observed association between differential gene expression and changes in DNA methylation might reflect functional local heterogeneity in chromosomal accessibility, which could involve certain structural changes in chromatin segments. The notion of isoDMBs may be useful to elucidate such heterogeneity or structural changes as a heuristic guide in future studies.” (lines 393-397).

Overall this study has yielded several important observations including well supported documentation of another example of transgenerational epigenetic transmission of defects/alterations in the epigenome follow exposure of a pregnant female to an endocrine disruptor, detailed assessments of the phenotypic effects persisting over at least 3 generations with no additional exposure to the disruptive agent following the initial exposure of the F0 generation pregnant dam, and extensive genomics level analyses of gene expression and the methylomes. Novel relationships between these parameters are suggested, which, with some clarification could represent a useful contribution to the field.

We thank the reviewer and hope that the clarifications we have made in the text have increased enthusiasm for our study.

Reviewer #2 (Remarks to the Author):

SUMMARY

The authors assess changes in DNA methylomes and transcriptomes in gonadal adipose tissue of mice ancestrally exposed to tributyltin (TBT). Rather than finding specific DNA methylation changes in regulatory regions, however, the authors identify larger chromosomal segments that are coordinately hypo- or hypermethylated (iso-DMBs) that may alter regional chromatin landscapes and, subsequently, gene expression. Differentially expressed genes within these iso-DMBs appear to be enriched for muscle biology and metabolism, suggesting these chromosomal alterations may be associated with the obese, leptin-resistant phenotype in ancestrally exposed mice.

MAJOR CONCERNS (GENERAL AND BY FIGURE).

- The authors make substantial claims that would be more convincing if additional evidence was provided. For example, other assessments of chromatin landscape (ex. histone modifications, chromatin accessibility, etc.) should be performed.

Specific examples:

1. regional changes in chromatin properties are reflected by changes in DNA methylation
2. TBT alters chromosome segmental structures that then result in changes in global DNA methylation

We agree with the reviewer that additional evidence for the substantial claims we made in the manuscript would be more convincing. The two points noted by the reviewer are our preferred interpretation of the data presented in our manuscript. Because the genomic endpoints we planned to analyze at the outset of the experiment were DNA methylation and transcriptomal changes, we used the available material for these analyses. Therefore, when it became clear that DNA methylation was not changing at specific promoters, but rather in isoDMBs, we did not have any freshly-prepared adipocyte nuclei available to analyze by ATAC-seq or Hi-C and related analyses as the reviewer is requesting.

However, we did have F3 and F4 sperm samples available from ancestrally TBT or vehicle treated animals and analyzed these by ATAC-seq in an attempt to provide direct evidence for transgenerational changes in genome accessibility. ATAC-seq analysis revealed that chromatin accessibility patterns differ between TBT and vehicle samples when guided by DNA sequence composition (using isochores as a proxy for sequence composition). Although, this is not exactly what the reviewer requested, we believe that these results establish an independent connection between ancestral TBT treatment and altered chromatin accessibility that may be indicative of changes in higher order chromatin organization that were transmissible through meiosis and mitosis. Moreover, F3 sperm generated the F4 generation of animals that we studied in this report and the effects noted in F4 sperm indicate that changes will likely be transmitted to the next generation. This is discussed in the text (lines 305-330).

The concept of an iso-DMB is difficult to understand with respect to the mechanism of ancestral TBT exposure. This is not really addressed in the paper. In the discussion the authors include one sentence relevant to this question: "The fact that

hyper and hypomethylated isoSMBs tend to show opposite biases for gene density and base composition suggests that TBT-dependent DNA methylation variation could be related to the spatial dynamics of the genome.”

3. What does this mean?

This sentence was rewritten to be more explicit about the relationship between TBT-dependent changes in DNA methylation and higher order chromatin organization. The revised section of text reads as follows: "Principles that differentiate specific regions of the genome seem to coalesce in two major genomic components that are recapitulated here by the opposite biases for gene density and base composition for hyper and hypomethylated isoDMBs (Fig. 2). We hypothesize that TBT-dependent changes in DNA methylation may not be the principle cause for the transgenerational inheritance of the ancestral exposure to TBT. Rather they are secondary to or caused by changes in higher order chromatin organization. Furthermore, the incipient association here detected between TBT-dependent changes in sperm chromatin accessibility and base composition (Fig. 5) is consistent with this hypothesis and also suggests that TBT-dependent alteration of higher order chromatin organization is transmissible through meiosis and mitosis. Further studies specifically designed to analyze the nuclear architecture of somatic and germ cells will be required to address specifically how TBT results in a transmissible alteration of the nuclear architecture." (Line 374-385). Our interpretation is that isoDMBs reflect TBT-dependent regional changes in higher order chromatin organization, and that the methylation changes themselves are secondary to the alteration of the higher order chromatin organization. The new section reporting on ATAC-seq results is consistent with the existence of transmissible TBT-dependent changes in higher order chromatin organization (lines 305-330).

4. Additionally, have these domains been independently defined in other studies, for example Hi-C studies?

Unambiguous, direct evidence such as Hi-C is not currently available for the linkage between isoDMBs and altered large-scale chromatin structure. We have moderated the statement to the following: “It is tempting to speculate that the observed association between differential gene expression and changes in DNA methylation might reflect functional local heterogeneity in chromosomal accessibility, which could involve certain structural changes in chromatin segments. The isoDMBs model may be useful to elucidate such heterogeneity or structural changes as a heuristic guide in future studies." (lines 393-397).

5. Are these blocks reflective of open and closed chromatin? This could easily be determined with ATAC-seq.

Due to the lack of material to perform Hi-C or ATAC-seq on adipose nuclei we could not address this directly in fat. However, the new section reporting on ATAC-seq analyses of sperm samples indicates that TBT causes local changes in DNA accessibility that we hypothesize may also reflect changes in higher order chromatin organization (lines 305-330).

Lack of rationale:

6. What was the rationale for looking at fat depot weights in F2-F4 generation animals when there were no significant differences in depot weight in the F1 generation?

We previously published a transgenerational study³ (independent from the study reported here), in which we found that F1 animals *in utero* exposed to TBT did not show increased epididymal fat depot weight. Animals from F2 and F3 generations did show an increase in epididymal depot size suggesting a non-Mendelian mechanism on the transmission of the genomic information. In the current study, we 1) replicated the previously published study, 2)

expanded the analysis to the F4 generation, 3) showed that ancestral obesogen exposure altered the response to diet in the F4 generation and propose a plausible framework for how endocrine disrupting chemicals such as TBT might act to promote transgenerational effects. Although, the number of labs performing transgenerational experiments has grown in the last few years, we are among the only ones to have reproduced the effects of EDC exposure at environmentally relevant doses in multiple, independent experiments.

7. Why was the methylome and transcriptome of gonadal fat assessed over other fat depots? Is it known whether different fat pads are more/less susceptible to environmental stimuli such as endocrine disruptors?

White adipose depots are broadly divided into two main groups, visceral (abdominal) and subcutaneous fat depots. We previously showed that TBT has a transgenerational effect on both visceral and subcutaneous fat depot size.³ Michael Skinner (Washington State University) also reported increased abdominal fat but not subcutaneous fat in rats after ancestral exposure to very high doses of endocrine disrupting chemicals such as BPA, DEHP, DBP, DTT and hydrocarbons. Vom Saal and colleagues have identified depot-specific responses to bisphenol A in F1 mice.⁴ Visceral and subcutaneous fat are functionally and structurally different. Increased visceral fat is directly associated with insulin resistance and cardiovascular risk; whereas subcutaneous fat is known as a healthier component of the total body fat.⁵ Due to the large number of animals and endpoints in this study, we chose to evaluate total body fat by EchoMRI and to collect only a single visceral depot for subsequent analysis.

Lack of validation:

1. The most important validation needs to be on the DNA methylation. The assay used here IPs methylated DNA in a nonlinear fashion (regions with more methylated CpGs density are greatly favored).

We appreciate the reviewer for raising this question about the importance of discussing technical aspects of the precipitation-dependent approaches to DNA methylation analysis.

The method adopted in our present study is MBD-seq, which uses the recombinant methylcytosine binding domain of human MBD2 for enrichment of methylated DNA fragments. Although the reviewer commented, “The assay used here IPs methylated DNA...,” MBD-seq does not ImmunoPrecipitate methylated DNA fragments. This is in contrast to MeDIP-seq, in which methylated DNA fragments are immunoprecipitated by anti-5-methylcytosine antibodies. Although these methods are conceptually similar, the outcomes of MBD-seq and MeDIP-seq are not identical, as demonstrated by Harris et al.⁶ and in a recent study of Neary et al.⁷

The outcomes of both MeDIP-seq and MBD-seq are affected by local density of CpG dinucleotides.⁸ Chavez et al. addressed this issue by normalizing MeDIP/MBD-seq deep sequencing data using local CpG density in mammalian genomes.⁹ Chavez’s method generates a relative methylation score (RMS) which serves as a CpG density-normalized indicator of DNA methylation. The Bioconductor package “MEDIPS”, which we used for MBD-seq data analysis in the present study was developed Chavez’s laboratory specifically for MeDIP/MBD-seq data analysis using RMS.¹⁰ Thus the effects of CpG density on MBD-seq DNA methylation analysis are reasonably adjusted in our study.

It is important to note that even if RMS of MBD-seq data analysis outputs could still have a bias between low- and high-CpG density areas within the mouse genome, such bias would

not significantly affect our identification of the TBT-affected Differentially Methylated Regions (DMRs). This is because DMRs are determined by comparing RMS between the same genomic DNA regions of animals with or without ancestral exposure to TBT.

The DMRs identified in our present study typically span 0.5 kbp – 2 kbp in length (Figs. 4B, S7B, S8). These DMRs appear as “peaks” involving gradual changes in DNA methylation of multiple 100-bp windows before and after a summit of each DMR peak. It is important to note that RMSs of each window as well as the DMRs reflect the overall tendency of altered DNA methylation of the respective regions – rather than strong or weak methylation of a small number of CpG sites. For this reason, conventional bisulfite pyrosequencing approaches, which can interrogate only 100-150 base pair regions of genomic DNA and have significant restrictions based on PCR and sequencing primer design, are not a practical method to validate genomic DNA methylation profiles determined by MBD-seq. Apparently whole-genome bisulfite sequencing (BS-seq) is the only appropriate approach to “validate” the genome-wide DNA methylation profiling performed using MBD-seq. However, this is not a practical strategy for the present study because each whole-genome bisulfite sequencing will require 400 million or greater numbers of deep sequencing reads for adequate quantitation of CpG methylation. Reduced representation bisulfite sequencing (RRBS) is also inadequate due to its extreme bias toward promoter-associated CpG islands, which we did not find in our analysis.

For all of these reasons, direct validation of DMRs determined by our genome-wide MBD-seq turned out to be technically challenging. Considering that we used n=4 for each genome-wide MBD-seq analysis with statistically rigorous analysis, and that we are not claiming methylation changes at specific nucleotides, but rather in regions, we believe that validation of methylation at specific sites in the genome is not warranted or feasible for this study

2. Leptin mRNA levels in additional animals (besides ones used for RNA-Seq)?

We performed qPCR analyses of adipose tissue from 8 animals from each group (control and treatment) including the samples analyzed by RNAseq. We found that there was an increase in the mRNA levels in the TBT group (~1.56 fold). However, due to high variability, this difference was not significant. Since the functional molecule is leptin protein, we analyzed plasma leptin levels in all the males studied in this experiment by ELISA (10 animals from DMSO group and 12 animals from TBT group) as an independent way of validation. We found that the protein levels are significantly increased, which supports our inference of a leptin resistant phenotype. These data can be found in Supplementary Table 3 and the description of the results were included in lines 146-159.

3. If circulating/serum leptin is also increased in ancestrally TBT-exposed animals, would leptin mRNA levels increased in other fat depots as well?

We thank the reviewer for this suggestion. Whether distinct fat depots equally respond to the ancestral TBT exposure or they show heterogeneous sensitivity is an interesting question to be addressed in future studies. We do not have tissue samples with which to measure leptin mRNA in other fat depots.

4. Are leptin levels significantly different in ancestrally TBT-exposed females compared to control females??

We analyzed leptin levels in plasma from females and these data are presented in Supplementary Table 3 and lines 146-159.

Figure 1

- Is this the F4 generation? This information should be included in the legend. Yes. The figure legend was modified to indicate that the data shown was obtained from F4 animals.

- Panels A and B: Body weight, % lean, and % fat mass: Animals in the ancestral TBT exposure group already displayed significantly increased percent body fat before the high fat diet challenge. Is the diet challenge necessary to see a significant difference in percent body fat over time between control and TBT groups?

Since we previously showed a transgenerational effect on fat depot size, we wanted to go even further and test whether ancestral TBT exposure elicited permanent changes in the response to diet. As the reviewer notes, there are already significant differences in overall fat mass at 19 weeks at the beginning of the diet challenge; therefore, the diet challenge is not strictly required to see significant changes in percent body fat over time in ancestrally TBT-treated animals vs. controls. However, as the results clearly demonstrate, ancestral TBT exposure results in an exacerbated response to diet and to fasting compared with controls. We believe that this result is highly pertinent to the study of obesity in humans.

- Also, is it known if all fat depots are equally affected by the dietary challenge? Our study evaluated total body fat mass and the weights of a single depot (gonadal). However, the total changes in fat mass observed are not accounted for by altered weights of this depot alone. Therefore, other we infer that the weights of other depots must also have been increased after the diet challenge. We cannot draw any conclusions about whether the depot weights were affected equally since only the gonadal depot was weighed in this experiment. However, in our previous publication where all depots were weighed, we did not observe substantially different responses in one depot but not in another.

- Panel D: There is a discordance between the figure legend and the panel? The figure legend states that plasma leptin is being measured at 8 and 33 weeks, before and after diet challenge. Main body of text (lines 133-136) makes it seem like these are plasma leptin levels measured in 33 week old animals ONLY that were subject to the HFD challenge.

We thank the reviewer for pointing out this error and modified the figure legend to reflect the data shown in the figure.

Figure 2

- If isoDMBs alter the entire chromosomal region, would you expect more genes to be differentially expressed within the isoDMB? The supplementary figures show that there are many genes unchanged within an isoDMB (especially sup Fig. 8). Also there are 2 under expressed genes and 3 overexpressed genes in this figure. Are the depicted regions the best examples?

Our interpretation is that TBT-dependent changes in higher order chromatin organization, reflected by isoDMBs, modulate DNA accessibility. Altered DNA accessibility is often sufficient to impose a general trend for gene expression, but there will also be cases in which the expression of a specific gene does not conform fully to the expected trend because other

elements important for regulation of gene expression (such as transcription factors, and co-regulators) may be absent or insufficient. We included a sentence to explain this point further (line 282-289). With regards to Supplemental figure 8, we agree with the reviewer's comment that this figure may lead to confusion. Therefore, we decided to remove it from the manuscript.

- It is unclear what the isochore data adds to this story. Does the associated isochore categorization contribute to predictability of isoDMB directionality (hypo- or hypermethylated) and/or gene expression (up- or downregulated)?

In this instance, we are using isochores as convenient proxies for differences in DNA sequence composition because the 5 isochores are defined by %GC. Therefore, including isochores on the figure directly compares isoDMBs with local DNA sequence composition. Isochores have recently been shown to correlate with higher order chromatin organization, specifically topologically associated domains and lamina associated domains.² A recent Hi-C study published in Nature,¹¹ showed that it was only possible to characterize higher order chromatin organization in single oocyte nuclei when Hi-C domains were mapped to DNA sequence composition. We included discussion of these papers in the revised manuscript (lines 266-270, and 317-319).

Figure 4

- Panel A: Should tick marks be pointing down if this is a hypomethylated isoDMB associated with overexpression of genes?

We agree with the reviewer. The figure was fixed

Supplementary Figure 8

- Iso-DMB is hypo- or hypermethylated? Make sure figure legend and image depict same directionality

Since Supplementary Figure 8 seems to be leading to confusion, we decided to remove that figure from the manuscript. .

Supplementary Table 1 and 2: Total weight and fraction of fat depots in F1-F4 males and females at 8 weeks of age

- Was this analysis done for the 33 week old animals?

No. We did not weigh the fat depots at 33 weeks of age because we wished to maximize tissue preservation during the dissection process for genomic analysis. Since this is the second time we are showing the transgenerational effects of TBT exposure on obesity, we decided to report the data at 8 weeks old to demonstrate the reproducibility of the experiment.

- Is the amount of variability across generations concerning?

We do not find the variation across generations concerning. Considering the experimental design in which F1 and F2 generation have been likely directly exposed (F1, during *in utero* development and F2 as the developing germ line of the F1) and F3 and F4 have not been exposed to TBT at any time, it is reasonable to expect that there may be some differences between the F1/F2, where there will be direct as well as epigenetic effects, and F3 and beyond where only epigenetic effects are possible.

- Is the severity of the phenotype diminishing over time?

The reviewer is right in that there is a very strong phenotype in F3 males, which is not as evident in F4, although F4 males still accumulate more gonadal fat than DMSO. After

challenging the animals with the higher fat diet, F4 TBT males accumulate a significantly increased amount of fat than controls suggesting that the animals may need to be metabolically challenged (e.g. fasting or increased fat diet) to make the phenotype more evident, which suggests that even though at 8 weeks of age the animals did not show a strong phenotype, but they were "poised" to gain weight if exposed to a diet challenge.

- Why/how do you explain the significant REDUCTION in inguinal fat mass in the fourth generation? This is not addressed at all in the body of the text.

Inguinal adipose tissue is part of the subcutaneous adipose depot. As indicated above, visceral and subcutaneous fat are functionally and structurally different⁵. The reduction of subcutaneous adipose tissue is associated with a reduction in insulin sensitivity, which is another marker of metabolic disruption. Therefore, the fact that these animals have increased visceral and decreased subcutaneous adipose depots may contribute to the negative response to metabolic inputs. We introduced a sentence in the text that addresses this point (lines 117-119)

MINOR CONCERNS.

The following concerns have all been addressed in the new version of the manuscript.

- Line 39: one's (need apostrophe) instead of ones.
- Line 42: squared symbol should be superscript in 2.3 kg/m²
- Making sure italicize in utero, in vitro, in vivo, ad libitum throughout the text (some examples/locations/lines: 43-44, 59, 62, 93, 95, 126, 248)
- Repetitive information (lines 170-185 and lines 435-451) classifying genes into Subsets I-III
- How old were the mice used for the methylome and transcriptome studies? Indicate this in the Methods

We thank the reviewer for these thoughtful comments and hope that the clarifications we have made in the text have increased enthusiasm for our study.

References:

1. Even-Faitelson, L., Hassan-Zadeh, V., Baghestani, Z. & Bazett-Jones, D.P. Coming to terms with chromatin structure. *Chromosoma* **125**, 95-110 (2016).
2. Jabbari, K. & Bernardi, G. An Isochore Framework Underlies Chromatin Architecture. *PLoS One* **12**, e0168023 (2017).
3. Chamorro-Garcia, R. *et al.* Transgenerational inheritance of increased fat depot size, stem cell reprogramming, and hepatic steatosis elicited by prenatal exposure to the obesogen tributyltin in mice. *Environ Health Perspect* **121**, 359-66 (2013).
4. Angle, B.M. *et al.* Metabolic disruption in male mice due to fetal exposure to low but not high doses of bisphenol A (BPA): evidence for effects on body weight, food intake, adipocytes, leptin, adiponectin, insulin and glucose regulation. *Reprod Toxicol* **42**, 256-68 (2013).
5. Ibrahim, M.M. Subcutaneous and visceral adipose tissue: structural and functional differences. *Obes Rev* **11**, 11-8 (2010).

6. Harris, R.A. *et al.* Comparison of sequencing-based methods to profile DNA methylation and identification of monoallelic epigenetic modifications. *Nat Biotechnol* **28**, 1097-105 (2010).
7. Neary, J.L., Perez, S.M., Peterson, K., Lodge, D.J. & Carless, M.A. Comparative analysis of MBD-seq and MeDIP-seq and estimation of gene expression changes in a rodent model of schizophrenia. *Genomics* (2017).
8. Down, T.A. *et al.* A Bayesian deconvolution strategy for immunoprecipitation-based DNA methylome analysis. *Nat Biotechnol* **26**, 779-85 (2008).
9. Chavez, L. *et al.* Computational analysis of genome-wide DNA methylation during the differentiation of human embryonic stem cells along the endodermal lineage. *Genome Res* **20**, 1441-50 (2010).
10. Lienhard, M., Grimm, C., Morkel, M., Herwig, R. & Chavez, L. MEDIPS: genome-wide differential coverage analysis of sequencing data derived from DNA enrichment experiments. *Bioinformatics* **30**, 284-6 (2014).
11. Flyamer, I.M. *et al.* Single-nucleus Hi-C reveals unique chromatin reorganization at oocyte-to-zygote transition. *Nature* **544**, 110-114 (2017).

Reviewers' comments:

Reviewer #1 (Remarks to the Author):

With this revised version of this manuscript, the authors have satisfactorily addressed the concerns raised regarding the original version and, in all but two cases, made appropriate changes to the text in the revision to clarify these points. However, in two cases, it appears that while a reasonable response to the reviewer's concern was provided in the authors' response to the reviewers in each case, similar information is still not conveyed within the text of the revised manuscript, thus failing to clarify these points for other readers. It is suggested that some modification be made to the text to incorporate the essence of the responses to these reviewer's concerns within the text of the revision. The two concerns in question are --

- Is a DMR that is not located near the TSS or promoter of such a gene considered functionally significant? It may well be that considerations such as these are, in fact, accommodated in the author's notion of an isoDMB, but if so, this needs to be explained more clearly in this reviewer's opinion.

and

- The discussion of potential effects at the level of "chromosome segmental structure" is both highly speculative and somewhat vague. Is this segmental structure similar to what others now refer to as topologically associated domains (TADs)?

As noted above, the authors' responses to these concerns are satisfactory, but those clarifications need to be included in the text of the revision as well.

Reviewer #2 (Remarks to the Author):

We thank the authors for their thoughtful response to the reviews. While we appreciate that tissues were limiting and the suggested chromatin experiments could not be done, most of the conceptual questions from the initial reviews still stand and it really isn't clear how ATAC-Seq of sperm addresses those questions. In fact, the abstract would lead the reader to think that DNA methylation fully explains the gene expression changes (abstract: "We found that ancestral TBT exposure induced global changes in DNA methylation together with altered expression of metabolism-relevant genes when the animals were exposed to dietary challenges.") Rather, the ATAC experiments address a completely different set of issues. In a sense, the authors have gone from examining the consequences of TBT treatment on the offspring to assessing how phenotypes can be transmitted. Both sets of questions are important and interesting but now both are superficially addressed in the revised manuscript.

That said, the new ATAC-Seq experiment is interesting and could potentially elevate the paper with additional analyses. For example.....

1. If transgenerational effects are occurring through epigenetic changes in the germline, could some changes in F3 sperm be conserved in F4 offspring adipose? Have you compared the amount of overlap between open chromatin from F3 sperm ATAC-Seq and the F4 son's hypomethylated isoDMBs in adipose tissue?

2. What proportion of the ATAC-Seq peaks in the TBT-treated F3 and F4 sperm overlap? Are any of the TBT-induced changes in chromatin accessibility associated with underlying metabolic genes? If so, could these regions and the affected genes be, at least partially, responsible for the transmission of the metabolic phenotype?

3. In Figure 5, there seem to be several F3 and F4 DMSO, and to a lesser extent TBT, sperm samples that cluster more closely together as opposed to stratifying by generation (i.e. F3 animals of the same treatment group clustering more closely to one another than F4 generation animals of the same treatment group). Are these father-son pairs (see highlighted in attachment)? Would father-son pairs be more likely to cluster than random F3 and F4 males? If so, can this be taken into account in the clustering analysis? That is, to what extent are these animals clustering by treatment vs. their shared lineage?

4. Minor comments:

- Lines 323-325: When referring to the differentiation of DMSO and TBT samples using hierarchical clustering of the isochores, you reference one isochore with GC content of 41-46%. I am assuming this is isochore H2? Could you explicitly state which isochore you are referring to? Or define isochores by their %GC content when you first introduce isochores beginning around Line 270.
- Supplementary information: Can the large tables (beginning at page 30, Supplementary Table 5) in the main text be put into Excel Files?

Reviewer #1 (Remarks to the Author):

With this revised version of this manuscript, the authors have satisfactorily addressed the concerns raised regarding the original version and, in all but two cases, made appropriate changes to the text in the revision to clarify these points. However, in two cases, it appears that while a reasonable response to the reviewer's concern was provided in the authors' response to the reviewers in each case, similar information is still not conveyed within the text of the revised manuscript, thus failing to clarify these points for other readers. It is suggested that some modification be made to the text to incorporate the essence of the responses to these reviewer's concerns within the text of the revision. The two concerns in question are --

We thank the reviewer for pointing two areas for which our previous responses had not been entirely reflected in the main text of the manuscript. We have addressed these two points in new revision of the manuscript as described below

- Is a DMR that is not located near the TSS or promoter of such a gene considered functionally significant? It may well be that considerations such as these are, in fact, accommodated in the author's notion of an isoDMB, but if so, this needs to be explained more clearly in this reviewer's opinion.

We have added the following text to the manuscript (lines 223-228) in response to the reviewer's comment. "Although isoDMBs do not preclude the possibility that an individual DMR could have short-range effects on the expression of genes that are close in cis or become close spatially through DNA looping, we hypothesize that isoDMBs reflect regional changes in higher order chromatin organization dependent on the ancestral exposure to TBT that alters the dynamics of DNA methylation and expression of genes encompassed in them."

and

- The discussion of potential effects at the level of "chromosome segmental structure" is both highly speculative and somewhat vague. Is this segmental structure similar to what others now refer to as topologically associated domains (TADs)?

Our previous response to the reviewer's concern is now more explicitly addressed in lines (418-422). "Moreover, regions with uniform base composition have been associated with higher order chromatin structures such as topological associated domains (TADs). Our results showing a significant overlap between F4 adipose isoDMBs and regions with uniform base composition underscore the possibility of a yet-to-be-confirmed association between TBT-dependent adipose isoDMBs and TADs."

As noted above, the authors' responses to these concerns are satisfactory, but those clarifications need to be included in the text of the revision as well.

Reviewer #2 (Remarks to the Author):

We thank the authors for their thoughtful response to the reviews. While we appreciate that tissues were limiting and the suggested chromatin experiments could not be done, most of the conceptual questions from the initial reviews still stand and it really isn't clear how ATAC-Seq of sperm addresses those questions. In fact, the abstract would lead the reader to think that DNA methylation fully explains the gene expression changes (abstract: "We found that ancestral TBT exposure induced global changes in DNA methylation together with altered expression of metabolism-relevant genes when the animals were exposed to dietary challenges.") Rather, the ATAC experiments address a completely different set of issues. In a sense, the authors have gone from examining the consequences of TBT treatment on the offspring to assessing how phenotypes can be transmitted. Both sets of questions are important and interesting but now both are superficially addressed in the revised manuscript.

We thank the reviewer for these helpful comments. We rewrote the abstract sentence highlighted by the reviewer, and expanded our analysis of ATAC-seq samples to better characterize the effects of ancestral TBT exposure on F3 and F4 sperm chromatin accessibility, taking care to address the specific issues raised by this reviewer.

That said, the new ATAC-Seq experiment is interesting and could potentially elevate the paper with additional analyses. For example.....

1. If transgenerational effects are occurring through epigenetic changes in the germline, could some changes in F3 sperm be conserved in F4 offspring adipose? Have you compared the amount of overlap between open chromatin from F3 sperm ATAC-Seq and the F4 son's hypomethylated isoDMBs in adipose tissue?

In the revised version of this manuscript, we performed additional analyses of the ATAC-seq data sets as suggested by the reviewer. This analysis revealed that there is indeed significant overlap between chromatin accessibility in F3 sperm revealed by ATAC-seq and isoDMBs in F4 adipose tissue from ancestrally TBT-treated animals compared with controls (but as noted below experimental design did not allow breeding of the F3 males analyzed to generate F4 sons). These data are now presented in Figure 5 and Supplementary Figure 12. Our analysis show that reductions in the ATAC-seq peaks (reflecting more "closed" chromatin) of F3/F4 sperm significantly overlaps with hypomethylated isoDMBs in F4 fat in the TBT group. Concomitantly, increases in the ATAC-seq peaks (reflecting more "open" chromatin) of F3/F4 sperm significantly overlap with hypermethylated isoDMBs in F4 fat in the TBT group. A new transgenerational study would be needed to offer a more complete interpretation for the continuity in TBT-dependent epigenetic changes that explain the transgenerational inheritance of TBT exposure. However, we offer one possible interpretation (albeit a speculation), that more closed chromatin in sperm could diminish post-fertilization transcription and/or epigenetic reprogramming, resulting in hypomethylation-associated isoDMBs (Lines 435-440). It is important to point out that changes in epigenetic marks observed in sperm may not necessarily be copied directly to the genomes of the phenotype-associated somatic cells. Germline epimutations may rather affect tissue- and sex-dependent epigenetic reprogramming in the genomes of somatic cells.

2. What proportion of the ATAC-Seq peaks in the TBT-treated F3 and F4 sperm overlap? Are any of the TBT-induced changes in chromatin accessibility associated with underlying metabolic genes? If so, could these regions and the affected genes be, at least partially, responsible for the transmission of the metabolic phenotype?

We performed the analyses proposed by the reviewer and included the results in lines 357-370 " to assess whether TBT-dependent changes in F3 and F4 sperm chromatin accessibility related with the metabolic phenotype observed in F4 animals, we functionally characterized genes spanned by DAIs in F3 and F4 sperm. First, we enquired about the functionality of genes spanned by DAIs found significant and with the same direction of change in both generations, i.e., F3-F4 shared DAIs (Supplementary Table 21). The fraction of F3-F4 shared DAIs is significantly larger than expected by chance (Fig. 6A), but genes spanned by them do not show any significant GO term enrichment (Supplementary Tables 21-22). Second, we look for significant enrichments for genes spanned by significant DAIs in the same direction for each generation separately. GO terms found enriched for genes spanned by the same type of DAIs in each generation showed a strong similarity (Fig. 6B and Supplementary Table 23). Particularly noticeable is the fact that all GO terms found enriched for both inaccessible F3 DAIs and inaccessible F4 DAIs are associated with metabolic functions (Fig. 6C and Supplementary Table 23), which connects with the metabolic disruption we observed for F4 animals ancestrally exposed to TBT."

3. In Figure 5, there seem to be several F3 and F4 DMSO, and to a lesser extent TBT, sperm samples that cluster more closely together as opposed to stratifying by generation (i.e. F3 animals of the same treatment group clustering more closely to one another than F4 generation animals of the same treatment group). Are these father-son pairs (see highlighted in attachment)? Would father-son pairs be more likely to cluster than random F3 and F4 males? If so, can this be taken into account in the clustering analysis? That is, to what extent are these animals clustering by treatment vs. their shared lineage?

In our experimental set up, the animals sacrificed in F3 are not used for breeding to contribute to subsequent generations, which means that there are no father-sons pairs in this analysis and no possibility of animals clustering by shared lineage rather than treatment. One reason why we choose this experimental design is that mating is a highly energy demanding activity, which can significantly alter the fat/lean balance (generally mated males are leaner than naive males from the same litter). Additionally, mating can also alter hormone levels, which could also be a confounding factor in our experiment.

4. Minor comments:

- Lines 323-325: When referring to the differentiation of DMSO and TBT samples using hierarchical clustering of the isochores, you reference one isochore with GC content of 41-46%. I am assuming this is isochore H2? Could you explicitly state which isochore you are referring to? Or define isochores by their %GC content when you first introduce isochores beginning around Line 270.

The ATAC-seq section in the results has been rewritten and we included the modification suggested by the reviewer.

- Supplementary information: Can the large tables (beginning at page 30, Supplementary Table 5) in the main text be put into Excel Files?

All the large tables will be provided as Excel files.

REVIEWERS' COMMENTS:

Reviewer #2 (Remarks to the Author):

The authors have addressed the request to analyze the ATAC data in greater detail.